# ConRep4CO: Contrastive Representation Learning of Combinatorial Optimization Instances across Types

**Ziao Guo**[1], **Yang Li**[1,2], **Shiyue Wang**[2,3], **Junchi Yan**[1,2] *

[1]School of Computer Science & School of Artificial Intelligence, Shanghai Jiao Tong University
[2]Shanghai Innovation Institute
[3]School of Mathematical Sciences, Key Laboratory of MEA,
and Shanghai Key Laboratory of PMMP, East China Normal University
{ziao.guo,yanglily,yanjunchi}@sjtu.edu.cn  wangshiyue@stu.ecnu.edu.cn

## ABSTRACT

Considerable efforts have been devoted to machine learning (ML) for combinatorial optimization (CO) problems, especially on graphs. Compared to the active and well-established research for representation learning of text and vision, etc., it remains under-studied for the representation learning of CO problems, especially across different types. In this paper, we try to fill this gap (especially for NP-complete (NPC) problems, as they, in fact, can be reduced to one another). Our so-called ConRep4CO framework, performs contrastive learning by first transforming CO instances in various original forms into the form of Boolean satisfiability (SAT). This scheme is readily doable, especially for NPC problems, including those practical graph decision problems (GDPs) which are inherently related to their NP-hard optimization versions. Specifically, each positive pair of instances for contrasting consists of an instance in its original form and its corresponding transformed SAT form, while the negative samples are other instances not in correspondence. Extensive experiments on seven GDPs (most of which are NPC) show that ConRep4CO significantly improves the representation quality and generalizability to problem scale. Furthermore, we conduct extensive experiments on NP-hard optimization versions of the GDPs, including MVC, MIS, MC and MDS. The results show that introducing ConRep4CO can yield performance improvements of **61.27%**, **32.20%**, **36.46%**, and **45.29%** in objective value gaps compared to problem-specific baselines, highlighting the potential of ConRep4CO as a unified pre-training paradigm for CO problems. Source code is available: https://github.com/Thinklab-SJTU/ConRep4CO.

## 1 INTRODUCTION

Combinatorial optimization (CO) has been attracting wide interest for its practical importance from logistics [49] to finance [44]. Compared with the vector or matrix-like data, e.g., image, text, and the associated short-range tasks, e.g., classification and regression, the CO problems are inherently more challenging due to their discrete and non-convex nature with complex constraints, which often leads to NP-complete (NPC) or even NP-hard complexity [29].

Despite the recent extensive research on machine learning (ML) for CO [3; 17; 63], there still exist many limitations: compared to the well-developed learning approaches for representation of text [43] and vision [58], the tailored representation learning framework for CO problems remains under-explored. In fact, existing ML4CO literature in technique is mainly tailored to a single problem type, e.g., TSP [51], graph matching [59; 62], which may become a bottleneck for their ability in the sense of not leveraging the potential cross-domain learning. This gap is especially pronounced with the fast

*Correspondence author. This work was partly supported by National Natural Science Foundation of China (92370201, 625B2119) and Fundamental and Interdisciplinary Disciplines Breakthrough Plan of the Ministry of Education of China, JYB2025XDXM411.

development of multi-modality joint representation learning out of the CO area, e.g. CLIP [46] for both text and image.

In this paper, we try to fill the above gap by advancing the pre-training representation learning paradigm for CO problems, particularly by selecting various NPC problems, mainly including the so-called graph decision problems (GDPs), as pre-training tasks [1]. The hope is that the model trained jointly on various problem types will exhibit better expressiveness and generalization ability. To achieve our objective, we particularly leverage an important fact: NPC problems can be reduced to one another, and can all be transformed into Boolean satisfiability (SAT), making SAT a common form to bridge different original forms. This also suggests that they inherently share a latent structure worth being exploited for effective representations and ultimately for the goal of high-quality problem solving. Furthermore, there is a strong connection between GDPs and their optimization versions, which are more practical in real-world applications. Most CO problems can be converted into their decision versions by adding a target value. Prior results [28] have also established polynomial-time equivalences between decision and optimization for key CO problems. We select GDPs as pretraining tasks to learn representations, maintaining the potential to leverage the learned representations to enhance general CO problem-solving beyond GDPs.

Specifically, we develop a contrastive pre-training learning paradigm, ConRep4CO, tailored to CO problems beyond the vanilla version for images [10]. The pre-training is performed by contrasting problem instances, where the positive sample is defined as a pair of a vanilla GDP instance with its corresponding SAT form, while the negative sample is a pair that is not in correspondence. Also, a decision loss is applied to guide each model to effectively learn the feature representations of the respective instances and capture the unique characteristics of its assigned problem domain. Extensive experiments are conducted to evaluate the effectiveness of the ConRep4CO paradigm, comprising two parts: **1) representation evaluation**, and **2) enhancement of CO problem-solving**. For 1), since there is no universal metric for CO representation evaluation, in the context of our discussion, we use the accuracy of solving GDPs as a measure. We first assess the representation quality and generalizability to problem scale by solving GDPs on both pre-training identical distribution and more difficult instances. We also evaluate the cross-domain generalizability by the solving performance on unseen GDP domains. For 2), we incorporate ConRep4CO into the training of problem-specific neural solvers for minimum vertex cover (MVC), maximum independent set (MIS), maximum clique (MC), and minimum dominating set (MDS). The neural solvers enhanced by ConRep4CO consistently show significant performance improvements, demonstrating the practical applicability of ConRep4CO beyond GDPs. **The highlights of the paper are as follows.**

1) We try to advance the frontier of representation learning, beyond the classic instance forms, e.g., text/vision, by proposing ConRep4CO, a novel contrastive pre-training paradigm to learn general representations across different CO problems with complex discrete constraints and variables.

2) We leverage the SAT form to build the positive/negative pairs for our carefully designed contrastive learning scheme, based on the fact that GDPs (i.e., NPC problems) can be reduced into the SAT form. A merit is that our contrastive approach is augmentation-free, as CO instance augmentation itself is a notoriously challenging task due to unique problem structures. This is in contrast to the trivial image augmentation as done in contrast to vision problems, which, in fact, is also a bottleneck for directly reusing the contrastive learning approaches in vision to combinatorial tasks.

3) Our method learns the representation across different types beyond a single type. Such a unified paradigm facilitates representation learning through knowledge transfer among problem domains and mutual enhancement. Extensive experiments show that ConRep4CO not only improves representation quality but also significantly enhances problem-solving for various CO problems.

## 2 RELATED WORK

**Machine Learning for CO.** The application of machine learning to graph-based CO problems has a rich history, with recent research demonstrating substantial advancements in this domain [30; 3; 41]. These problems span a wide range of domains, including classical combinatorial optimization tasks (e.g., MILP [20; 74], SAT [76; 34; 11]) as well as structured generation problems such as

---

[1]GDPs are the decision versions of general NP-hard CO problems, such as the $k$-independent set problem corresponding to the maximum independent set (MIS) problem, and encapsulate the core challenges of CO. From the 21 NP-complete problems identified by [29], 10 are GDPs, highlighting their fundamental importance.

molecule generation [71; 68] and protein docking [67]. Most ML-based approaches for CO follow a two-stage framework: *(1) Graph representation learning*, where graph instances are embedded into low-dimensional vector spaces through graph neural networks (GNNs) [23; 6; 8; 70; 39]; and *(2) The utilization of these learned representations to solve CO problems* [27; 45; 48; 61; 64; 38; 36; 35; 37; 9]. Our ConRep4CO paradigm focuses on enhancing the first stage by proposing a more general training approach. While previous work has largely focused on designing network architectures [31; 22; 54; 60], our approach emphasizes the development of a training paradigm that leverages information from multiple problem types. There are also recent works on training with different types of CO, e.g., GOAL [12], UniCO [42] and MAB-MTL [57]. However, these works are orthogonal to ours as they directly follow the multi-task paradigm without contrastive pre-training.

**Graph Contrastive Learning.** Many graph contrastive frameworks rely on graph augmentations, which can be broadly categorized into two types: *(1) structural perturbations*, such as node dropping, edge sampling, and graph diffusion [13; 26]; and *(2) feature perturbations*, such as adding noise to node features [24]. These augmentation strategies have demonstrated effectiveness across a range of tasks, from graph-level representations [24; 73] to node-level representations [56; 53]. Our ConRep4CO paradigm moves beyond traditional graph augmentations by contrasting graph instances across multiple problem types. Instead of solely relying on structural and feature perturbations, ConRep4CO leverages the inherent characteristics of different CO problems, enabling the model to capture higher-level characteristics. Note that our approach is augmentation-free. We believe this is a nice property as graph augmentation[2] itself is a notoriously complex problem and much more complex than that on image data, as done in the vanilla contrastive paradigm SimCLR [10], where the augmented positive samples are generated by adding perturbations on the image e.g,. cropping, translating, warping etc.

## 3 METHODOLOGY

We present details of our **Con**trastive **Rep**resentation alignment and learning **for C**ombinatorial **O**ptimization (**ConRep4CO**) paradigm. We start by introducing the preliminary background on representations of graph decision problems and SAT in Sec. 3.1. Then, we elaborate on our approach to aligning multiple problem types in Sec. 3.2. Finally, we introduce the overall pipeline and implementation of our ConRep4CO, as well as some important training details in Sec. 3.3.

### 3.1 PRELIMINARIES

**Graph decision problem (GDP).** As a fundamental computational challenge, its goal is to determine the existence of specific properties within a given graph. These properties can vary, from identifying whether a graph contains a particular substructure, e.g. a clique or cycle, to assessing whether it meets conditions like connectivity or planarity. Graph decision problems are typically the decision versions of general NP-hard CO problems, e.g., the $k$-independent set problem corresponding to the maximum independent set problem, and the $k$-vertex cover problem corresponding to the minimum vertex cover problem, making them essential in the context of NPC problems. In particular, ML-based models can be effectively utilized to address GDPs. The objective is to learn a representation of a specific GDP type and use it to predict decisions based on the input graph. These representations can be understood as mappings that translate the structural properties of the input graphs into corresponding decisions, thereby capturing the underlying patterns required for decision-making in GDPs.

**SAT problem.** A Boolean formula in propositional logic consists of Boolean variables connected by logical operators "and" ($\land$), "or" ($\lor$), and "not" ($\neg$). A literal, denoted as $l_i$, is defined as either a variable or its negation, and a clause $c_j$ is represented as a disjunction of $n$ literals, $\bigvee_{i=1}^{n} l_i$. A Boolean formula is in Conjunctive Normal Form (CNF) if it is expressed as a conjunction of clauses $\bigwedge_{j=1}^{m} c_j$. Given a CNF formula, the Boolean Satisfiability Problem (SAT) aims to determine whether there exists an assignment $\pi$ of Boolean values to its variables under which the formula evaluates to true. If such an assignment $\pi$ exists, the formula is called satisfiable, where $\pi$ is called a satisfying assignment; otherwise, it is unsatisfiable. Graph representations play an important role in analyzing SAT formulas, with four common primary forms [4]: the literal-clause graph (LCG), literal-incidence graph (LIG), variable-clause graph (VCG), and variable-incidence graph (VIG). The LCG is a bipartite graph consisting of two types of nodes—literals and clauses—where an edge between a literal and a clause signifies the occurrence of that literal in the clause. LIG, in contrast, consists

---

[2]The CO instance can often be represented as a certain graph, e.g., a bipartite graph in [16].

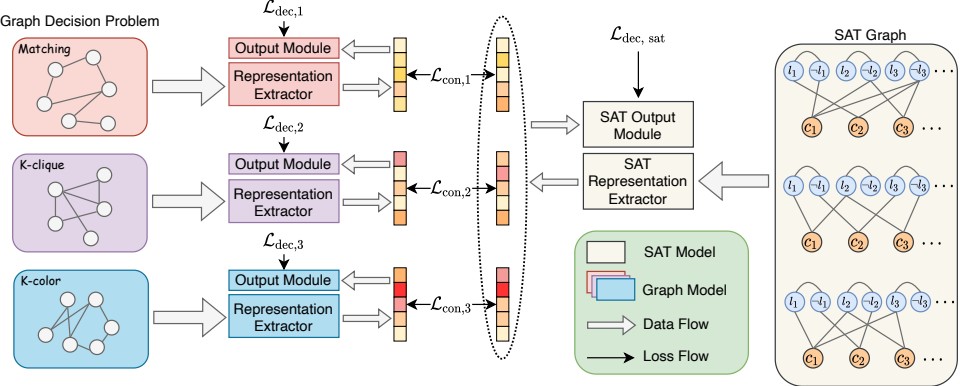

Figure 3: Overview of ConRep4CO with the proposed contrastive learning scheme. Given instances from multiple GDP types and their corresponding SAT graphs, a graph model is trained for each GDP type alongside a SAT model. Each model is composed of a Representation Extractor and an Output Module. The input graphs are processed by the Representation Extractor to generate instance-level representations, which are subsequently fed into the Output Module to produce the final decisions for each instance. The decision loss is applied individually to each model, while the contrastive loss is applied to each graph model. All contrastive losses are applied to the SAT model.

solely of literal nodes, with edges representing the co-occurrence of two literals within the same clause. VCG and VIG are derived from LCG and LIG by merging each literal with its negation.

## 3.2 MODAL ALIGNMENT

We aim to enhance the learned representations of graph instances across a diverse range of GDPs by incorporating and synthesizing information from multiple GDP types. Specifically, we conceptualize each GDP type as a distinct problem modality. By adopting this multi-modal perspective, we explore the potential for cross-modal information-passing schemes. Note that the term 'modality' is not strictly defined. We hope to express that the problems represent different forms of a higher-level underlying difficulty and share a common underlying structure.

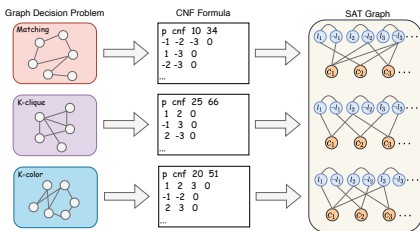

Figure 1: Transformation process from various GDP instances to the unified LCG representation of SAT.

Significant challenges arise due to the inherent disparities and structural gaps between different GDP types, often exhibiting varying graph topologies and problem characteristics. To fill this gap, we propose introducing SAT as a unified intermediary modality. The core concept involves transforming each GDP instance into its corresponding CNF formula, effectively converting it into a SAT instance. Once transformed, we construct a SAT-based graph representation for each instance, ensuring that all GDP instances, regardless of their original modalities, are standardized into an equivalent SAT graph representation. This transformation allows for uniform modeling across disparate problem types. Fig. 1 clarifies our approach to modal transformation.

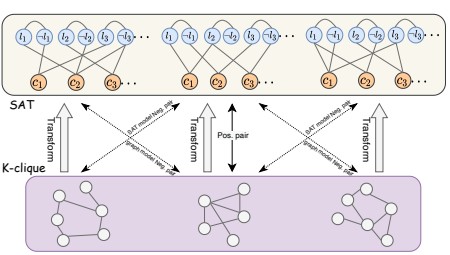

Figure 2: An example of the contrastive learning process, where 'Pos. pair' and 'Neg. pair' refer to positive and negative pairs, respectively. A similar process applies to other modalities with SAT.

After this transformation, we leverage contrastive learning to align the different modalities. Specifically, each GDP instance and its corresponding SAT instance form a positive pair to train both the SAT and graph models, while SAT instances derived from other GDP instances within the same GDP type serve as negative samples for the graph model. Similarly, other GDP instances within the same type serve as negative samples for the SAT model. Fig. 2 illustrates the contrastive learning process. The SAT modality, in turn, aligns with all other modalities. The goal is to facilitate information transfer across GDP modalities while preserving the distinct characteristics of each problem type.

### 3.3 CONREP4CO PARADIGM

#### 3.3.1 OVERVIEW

In this section, we provide a detailed introduction to ConRep4CO. Fig. 3 exhibits an overview.

Consider $n$ types of GDPs, denoted as $\mathcal{P}_1, \mathcal{P}_2, \ldots, \mathcal{P}_n$, along with $n$ corresponding graph sets $\mathbf{G}_1, \mathbf{G}_2, \ldots, \mathbf{G}_n$. For simplicity, assume that each graph set $\mathbf{G}_i$ contains $m$ graphs, i.e., $\mathbf{G}_i = \{\mathcal{G}_i^1, \mathcal{G}_i^2, \ldots, \mathcal{G}_i^m\}$, for $i = 1, 2, \ldots, n$. The objective is to solve problem $\mathcal{P}_i$ on graphs in $\mathbf{G}_i$. In total, there are $m \times n$ instances, denoted by $I_i^j = (\mathcal{P}_i, \mathcal{G}_i^j)$, where $i = 1, 2, \ldots, n$ and $j = 1, 2, \ldots, m$.

We first transform each of the $m \times n$ GDP instances into CNF, thereby generating their corresponding SAT graphs, i.e., $(\mathcal{P}_i, \mathcal{G}_i^j) \to \mathcal{B}_i^j$, where $\mathcal{B}_i^j$ is the constructed SAT graph.

Then, we develop $n$ distinct graph models, $\mathbb{M}_1, \ldots, \mathbb{M}_n$, each for one GDP type, and one unified SAT model $\mathbb{M}_{sat}$ to address the problem space. Both the graph models and the SAT model are structured around two key components: the **Representation Extractor** and the **Output Module**. The Representation Extractor, implemented as a GNN-based network, is responsible for learning and extracting representations from the input graph instances, whether derived from GDP or SAT transformations. The Output Module, implemented as an MLP, then utilizes these learned representations to produce task-specific outputs, thereby enabling the resolution of the given problem.

In training, we jointly train the $(n + 1)$ models corresponding to the $n$ GDP modalities along with the SAT modality. The supervision is derived from two parts: the decision loss and the contrastive loss. The decision loss is applied independently to each model, guiding the encoder to learn domain-specific feature representations. Meanwhile, contrastive loss is employed to facilitate feature fusion and message passing across different modalities, enabling the models to leverage complementary information from other problem domains. For training sample and label preparation, one cost comes from generating the corresponding SAT form for each instance, which requires a polynomial complexity. While this cost is negligible for computing labels for the training instances.

#### 3.3.2 LOSS FUNCTION

We introduce the definition and computation of the two key loss functions used in ConRep4CO.

**Decision Loss $\mathcal{L}_{\mathbf{dec}}$.** It is defined as a binary cross-entropy loss, which can be computed by:

$$\mathcal{L}_{\text{dec}} = \sum_{i \in \text{Batch}} \left\{ -d_i^{\text{gt}} \log(d_i^{\text{out}}) - (1 - d_i^{\text{gt}}) \log(1 - d_i^{\text{out}}) \right\}, \tag{1}$$

where $d^{\text{out}}$ denotes the output decision of the models, and $d^{\text{gt}}$ refers to the ground truth label. For each model, the decision loss is independently computed and applied.

**Contrastive Loss $\mathcal{L}_{\mathbf{con}}$.** Here we adopt the classic contrastive objective as widely used in literature [10; 25; 52], to facilitate the alignment between the GDP and SAT modalities. Taking $\mathcal{P}_n$ and the SAT modality as an example, $\mathcal{L}_{\text{con}}$ is formulated as:

$$\mathcal{L}_{\text{con},n} = \sum_{i=1}^{N} \left\{ -\log \frac{\exp(sim(\hat{\mathbf{r}}_n^i, \hat{\mathbf{r}}_{sat}^i)/\tau)}{\sum_{j=1}^{N} \mathbb{I}_{j \neq i} \exp(sim(\hat{\mathbf{r}}_n^i, \hat{\mathbf{r}}_{sat}^j)/\tau)} - \log \frac{\exp(sim(\hat{\mathbf{r}}_n^i, \hat{\mathbf{r}}_{sat}^i)/\tau)}{\sum_{j=1}^{N} \mathbb{I}_{j \neq i} \exp(sim(\hat{\mathbf{r}}_n^j, \hat{\mathbf{r}}_{sat}^i)/\tau)} \right\} \tag{2}$$

where $N$ represents the number of instance pairs in a batch, $\hat{\mathbf{r}}_n^i$ denotes the normalized representation of the $i$-th instance in the $\mathcal{P}_n$ modality, and $\hat{\mathbf{r}}_{sat}^i$ denotes the normalized representation of the corresponding instance in the SAT modality, derived from the $i$-th instance of the $\mathcal{P}_n$ modality. The parameter $\tau$ is the temperature scalar, and $\mathbb{I}$ is an indicator function. The function $sim(\cdot, \cdot)$ measures the cosine similarity between two representations, defined as $sim(\mathbf{r}_i, \mathbf{r}_j) = \frac{\mathbf{r}_i^\top \mathbf{r}_j}{\|\mathbf{r}_i\| \|\mathbf{r}_j\|}$.

Table 1: GDP solving accuracy (%) with confidence intervals ($\alpha = 0.05$) of the graph models trained on an identical distribution, measuring the quality of learned representations. 'SAT Back.' refers to SAT model backbone, and 'Graph Back.' denotes graph model backbone. The 'Overall' column represents the average accuracy across all datasets.

| SAT Back. | Graph Back. | Difficulty | Model | $k$-Clique | $k$-Domset | $k$-Vercov | $k$-Color | $k$-Indset | Matching | Automorph | Overall |
|---|---|---|---|---|---|---|---|---|---|---|---|
| LCG+NeuroSAT | GCN | Easy | Graph Model | 77.0±0.2 | 58.5±0.3 | 60.3±0.6 | 86.1±0.2 | 62.7±0.4 | 71.2±0.2 | 63.6±0.4 | 68.5 |
| | | | Graph Model+ConRep4CO | **79.3±0.3** | **62.0±0.1** | **67.3±0.2** | **90.2±0.1** | **67.5±0.5** | **71.7±0.3** | **65.4±0.3** | **71.9** |
| | | Medium | Graph Model | 63.2±0.5 | 62.2±0.2 | 59.9±0.4 | 79.6±0.2 | 61.1±0.2 | 70.6±0.5 | 63.3±0.4 | 65.7 |
| | | | Graph Model+ConRep4CO | **71.3±0.5** | **64.6±0.2** | **63.3±0.3** | **82.2±0.2** | **64.0±0.1** | **72.8±0.4** | **65.7±0.4** | **69.1** |
| LCG+GCN | GCN | Easy | Graph Model | 77.0±0.3 | 58.5±0.2 | 60.3±0.4 | 86.1±0.2 | 62.7±0.3 | **71.2±0.4** | 63.6±0.3 | 68.5 |
| | | | Graph Model+ConRep4CO | **79.3±0.2** | **61.1±0.3** | **65.0±0.3** | **89.6±0.1** | **67.7±0.4** | 71.1±0.3 | **64.6±0.2** | **71.2** |
| | | Medium | Graph Model | 63.2±0.4 | 62.2±0.2 | 59.9±0.3 | 79.6±0.3 | 61.1±0.2 | 70.6±0.4 | 63.3±0.3 | 65.7 |
| | | | Graph Model+ConRep4CO | **71.5±0.3** | **65.4±0.4** | **63.4±0.2** | **81.7±0.2** | **64.0±0.3** | **72.3±0.3** | **64.4±0.4** | **69.0** |
| VCG+GCN | GCN | Easy | Graph Model | 77.0±0.3 | 58.5±0.2 | 60.3±0.4 | 86.1±0.2 | 62.7±0.3 | **71.2±0.2** | 63.6±0.3 | 68.5 |
| | | | Graph Model+ConRep4CO | **78.0±0.4** | **60.6±0.3** | **62.9±0.3** | **88.8±0.1** | **66.3±0.2** | 71.1±0.3 | **64.2±0.2** | **70.3** |
| | | Medium | Graph Model | 63.2±0.4 | 62.2±0.2 | 59.9±0.3 | 79.6±0.3 | 61.1±0.2 | 70.6±0.4 | 63.3±0.3 | 65.7 |
| | | | Graph Model+ConRep4CO | **70.8±0.3** | **64.2±0.4** | **63.0±0.2** | **80.4±0.2** | **62.1±0.3** | **71.8±0.3** | **64.0±0.4** | **68.0** |
| LCG+NeuroSAT | GraphSAGE | Easy | Graph Model | 57.9±0.5 | 50.0±0.3 | 50.7±0.4 | 61.8±0.2 | 52.2±0.3 | 58.2±0.3 | 53.8±0.4 | 54.9 |
| | | | Graph Model+ConRep4CO | **79.7±0.2** | **63.2±0.3** | **70.8±0.3** | **93.3±0.1** | **75.3±0.4** | **71.0±0.2** | **63.9±0.3** | **73.9** |
| | | Medium | Graph Model | 52.8±0.4 | 56.5±0.2 | 56.0±0.3 | 55.2±0.3 | 50.0±0.4 | 58.2±0.3 | 54.8±0.2 | 54.8 |
| | | | Graph Model+ConRep4CO | **72.8±0.3** | **64.1±0.4** | **66.7±0.2** | **85.9±0.2** | **70.1±0.3** | **71.7±0.4** | **64.8±0.3** | **70.9** |
| LCG+NeuroSAT | PGN | Easy | Graph Model | 76.2±0.3 | 58.4±0.2 | 66.4±0.4 | 91.6±0.1 | 67.9±0.3 | 68.7±0.2 | 61.7±0.3 | 70.1 |
| | | | Graph Model+ConRep4CO | **77.3±0.2** | **61.9±0.3** | **69.7±0.3** | **93.7±0.2** | **71.6±0.4** | **70.3±0.3** | 61.7±0.2 | **72.3** |
| | | Medium | Graph Model | **72.4±0.4** | 62.8±0.3 | 64.7±0.2 | 83.0±0.3 | **68.1±0.2** | 58.8±0.4 | 50.4±0.3 | 65.7 |
| | | | Graph Model+ConRep4CO | 72.0±0.3 | **63.3±0.4** | **66.0±0.3** | **86.4±0.2** | 67.2±0.3 | **70.8±0.3** | **63.3±0.4** | **69.9** |
| LCG+NeuroSAT | GraphGPS | Easy | Graph Model | 82.4±0.2 | 77.2±0.3 | 85.5±0.1 | 89.9±0.2 | 76.4±0.3 | 69.4±0.4 | **67.4±0.2** | 78.3 |
| | | | Graph Model+ConRep4CO | **83.9±0.3** | **77.4±0.2** | **88.5±0.2** | **90.6±0.1** | **78.4±0.4** | **76.3±0.3** | 66.4±0.3 | **80.2** |
| | | Medium | Graph Model | 70.7±0.4 | 62.5±0.3 | 66.8±0.2 | 84.9±0.3 | 61.8±0.2 | **69.4±0.3** | 62.6±0.4 | 68.2 |
| | | | Graph Model+ConRep4CO | **71.7±0.3** | **72.9±0.4** | **81.8±0.3** | **85.6±0.2** | **73.0±0.3** | 57.2±0.4 | **63.2±0.3** | **72.2** |

Each GDP modality is trained using the contrastive loss with the SAT modality, allowing independent optimization for each GDP model. In parallel, the SAT model is optimized using the average contrastive losses computed across all GDP modalities, ensuring effective alignment.

### 3.3.3 TRAINING DETAILS

We adopt a warm start strategy to ensure the models learn robust representations. During the initial training phase, only the decision loss is utilized, while the contrastive loss is temporarily disabled. This phase allows the models to focus on learning meaningful task-specific representations based solely on the decision outcomes. Our insight is to provide a stable foundation for representation learning before introducing the more complex cross-modal alignment enforced by contrastive loss.

After the warm start phase, we introduce the contrastive loss alongside the decision loss. To balance the influence of these two losses, we introduce a parameter $\beta$, which controls the relative weight of the decision loss during the joint training phase.

### 3.3.4 INCORPORATING CONREP4CO INTO PROBLEM-SPECIFIC NEURAL SOLVERS

Despite selecting GDPs as the pre-training tasks, ConRep4CO is not restricted to only GDP solving in practical applications. ConRep4CO can be incorporated into problem-specific neural solvers to enhance the learned representation and ultimately improve problem-solving. Suppose one has pre-trained a SAT model (and graph models) using ConRep4CO. The following steps can be taken:

**1)** Convert the decision version instances of the neural solver's corresponding problem domain into CNFs. This can typically be implemented through off-the-shelf tools, such as CNFGen [33], and requires polynomial complexity, which is negligible for computing labels for the training instances.

**2)** Use the loss function defined in Sec. 3.3.2 to perform contrastive learning between the neural solver model and the pre-trained SAT model. The neural solver's architecture may need minor adjustments, such as adding an output module, while the parameters of the SAT model are fixed, as the pre-trained SAT model already contains unified representations for various problems.

**3)** Fine-tune the neural solver in the original problem domain to adapt the learned representations.

When incorporating ConRep4CO into different solvers, full retraining is unnecessary as the pre-trained SAT model is reusable, requiring only an efficient alignment procedure. Consequently, the need for training multiple models arises infrequently, significantly enhancing practical applicability.

Table 2: GDP solving accuracy (%) with confidence intervals ($\alpha = 0.05$) of the graph models on the hard datasets, measuring the generalizability of learned representations. 'SAT Back.' refers to SAT model backbone, and 'Graph Back.' denotes graph model backbone. The terms 'Easy' and 'Medium' in parentheses indicate the difficulty level of the datasets used for training. The 'Overall' column represents the average accuracy across all datasets.

| SAT Back. | Graph Back. | Model | $k$-Clique | $k$-Domset | $k$-Vercov | $k$-Color | $k$-Indset | Matching | Automorph | Overall |
|---|---|---|---|---|---|---|---|---|---|---|
| LCG+NeuroSAT | GCN | Graph Model (Easy) | 54.5±0.2 | 50.0±0.1 | 50.0±0.1 | 54.6±0.4 | 50.5±0.2 | 66.4±0.3 | 63.1±0.1 | 55.6 |
| | | Graph Model+ConRep4CO (Easy) | **57.1±0.1** | **50.1±0.1** | 50.0±0.1 | **60.5±0.3** | 50.3±0.2 | **67.9±0.4** | **63.6±0.2** | **57.1** |
| | | Graph Model (Medium) | 57.1±0.1 | 56.2±0.1 | 50.0±0.1 | 63.7±0.5 | 53.1±0.5 | 68.3±0.3 | 63.2±0.2 | 58.8 |
| | | Graph Model+ConRep4CO (Medium) | **57.8±0.2** | **56.5±0.1** | **57.7±0.3** | **67.6±0.5** | **56.5±0.4** | **70.0±0.2** | **65.3±0.2** | **61.6** |
| LCG+GCN | GCN | Graph Model (Easy) | **54.5±0.3** | 50.0±0.0 | 50.0±0.0 | 54.6±0.4 | **50.5±0.2** | 66.4±0.3 | 63.1±0.4 | 55.6 |
| | | Graph Model+ConRep4CO (Easy) | 52.5±0.4 | 50.0±0.0 | **53.9±0.3** | **55.7±0.2** | 49.9±0.3 | **68.6±0.2** | 63.1±0.3 | **56.2** |
| | | Graph Model (Medium) | 57.1±0.2 | 56.2±0.3 | 50.0±0.0 | 63.7±0.3 | 53.1±0.4 | 68.3±0.4 | 63.2±0.2 | 58.8 |
| | | Graph Model+ConRep4CO (Medium) | **57.9±0.3** | **58.9±0.2** | **57.4±0.4** | **65.6±0.3** | **55.2±0.3** | **71.2±0.3** | **64.5±0.4** | **61.5** |
| VCG+GCN | GCN | Graph Model (Easy) | **54.5±0.3** | 50.0±0.0 | 50.0±0.0 | 54.6±0.4 | **50.5±0.2** | 66.4±0.3 | 63.1±0.4 | 55.6 |
| | | Graph Model+ConRep4CO (Easy) | 53.1±0.2 | 50.0±0.0 | 50.0±0.0 | **55.4±0.3** | 49.6±0.4 | **68.4±0.4** | **63.4±0.3** | **55.7** |
| | | Graph Model (Medium) | 57.1±0.2 | 56.2±0.3 | 50.0±0.0 | 63.7±0.3 | 53.1±0.4 | 68.3±0.4 | 63.2±0.2 | 58.8 |
| | | Graph Model+ConRep4CO (Medium) | **57.7±0.4** | **60.5±0.3** | **57.7±0.3** | **64.8±0.2** | **53.6±0.3** | **69.0±0.3** | **64.3±0.3** | **61.1** |
| LCG+NeuroSAT | GraphSAGE | Graph Model (Easy) | 50.9±0.4 | 50.3±0.3 | 48.1±0.2 | 50.8±0.3 | 50.5±0.2 | 57.8±0.4 | 55.7±0.3 | 52.0 |
| | | Graph Model+ConRep4CO (Easy) | **52.9±0.3** | **59.9±0.4** | **55.9±0.4** | **60.2±0.2** | **58.5±0.3** | **67.9±0.3** | **62.1±0.4** | **59.6** |
| | | Graph Model (Medium) | 50.9±0.4 | 57.3±0.2 | 54.7±0.3 | 50.2±0.4 | 48.9±0.3 | 58.4±0.3 | 55.8±0.2 | 53.7 |
| | | Graph Model+ConRep4CO (Medium) | **59.7±0.3** | **59.5±0.3** | **60.3±0.2** | **70.2±0.4** | **56.4±0.4** | **68.4±0.2** | **64.2±0.3** | **62.7** |
| LCG+NeuroSAT | PGN | Graph Model (Easy) | 54.2±0.3 | 59.3±0.2 | 59.5±0.4 | 63.1±0.3 | 54.9±0.2 | 66.3±0.4 | 60.3±0.3 | 59.7 |
| | | Graph Model+ConRep4CO (Easy) | **54.6±0.2** | **59.8±0.3** | **59.9±0.3** | **63.3±0.4** | **55.1±0.3** | **66.7±0.3** | **61.0±0.4** | **60.1** |
| | | Graph Model (Medium) | 60.4±0.4 | 58.6±0.3 | 59.7±0.2 | 69.1±0.3 | 55.9±0.4 | 67.1±0.2 | **63.5±0.3** | 62.0 |
| | | Graph Model+ConRep4CO (Medium) | **61.2±0.3** | **58.9±0.4** | **60.7±0.3** | **69.7±0.2** | **58.1±0.3** | **67.5±0.3** | 63.3±0.2 | **62.8** |
| LCG+NeuroSAT | GraphGPS | Graph Model (Easy) | **59.6±0.2** | 50.0±0.0 | 49.9±0.1 | 50.0±0.0 | 53.5±0.3 | **68.0±0.4** | 57.6±0.3 | 55.5 |
| | | Graph Model+ConRep4CO (Easy) | 59.3±0.3 | **50.7±0.1** | **60.9±0.4** | **59.6±0.3** | 53.5±0.4 | 58.9±0.3 | **59.5±0.4** | **57.5** |
| | | Graph Model (Medium) | 63.2±0.3 | 55.2±0.4 | 56.8±0.2 | 68.3±0.4 | 63.0±0.3 | **63.9±0.3** | 58.3±0.4 | 61.2 |
| | | Graph Model+ConRep4CO (Medium) | **63.8±0.4** | **60.8±0.3** | **77.9±0.3** | **68.9±0.3** | **65.7±0.4** | 60.1±0.2 | **61.4±0.3** | **65.5** |

Table 3: GDP solving accuracy (%) with confidence intervals ($\alpha = 0.05$) of the graph models on Easy datasets. The 'Overall' column represents the average accuracy across all datasets.

| Model | $k$-Clique | $k$-Domset | $k$-Vercov | $k$-Color | $k$-Indset | Matching | Automorph | Overall |
|---|---|---|---|---|---|---|---|---|
| Graph Model | 76.2±0.2 | 58.4±0.4 | 66.4±0.2 | 91.6±0.5 | 67.9±0.3 | 68.7±0.3 | 61.7±0.2 | 70.1 |
| Graph Model-Unseen | **76.9±0.3** | **61.0±0.1** | **70.1±0.2** | **93.4±0.4** | **68.6±0.3** | **69.7±0.1** | 61.7±0.2 | **71.6** |

## 4 EXPERIMENTS

### 4.1 EXPERIMENTAL SETUP

**Datasets.** To evaluate the broad applicability of our approach, we select seven GDPs: $k$-Clique, $k$-Dominating Set ($k$-Domset), $k$-Vertex Cover ($k$-Vercov), $k$-Coloring ($k$-Color), $k$-Independent Set ($k$-Indset), Perfect Matching (Matching), and Graph Automorphism (Automorph). For each problem, we randomly generate graph instances that adhere to a distribution specific to the problem. To ensure a comprehensive and rigorous evaluation, we create datasets with varying levels of difficulty, categorized as easy, medium, and hard, based on the size and distribution of the generated graphs. For each easy and medium dataset, we generate 160,000 instances for training, 20,000 instances for validation, and 20,000 instances for testing. For each hard dataset, we only produce 20,000 instances for testing to evaluate the generalizability of the learned representations. Additionally, we ensure an equal distribution of labels, with 50% of instances labeled as satisfiable (1) and 50% as unsatisfiable (0) across the training, validation, and test sets. The graph instances were transformed into CNF using generators from CNFGen [33]. Furthermore, we evaluate the effectiveness of ConRep4CO in enhancing CO problem-solving on four practical CO problems: minimum vertex cover (MVC), maximum independent set (MIS), maximum clique (MC), and minimum dominating set (MDS). For MVC, we follow the setting in [72], using Erdős–Rényi (ER) graphs with three scales, containing approximately 50 to 100, 100 to 200, and 400 to 500 vertices, respectively. For MIS, MC, and MDS, we follow the setting in [75], using RB graphs [69] for MIS and MC, and BA graphs [2] for MDS, generating two scales of datasets with approximately 200 to 300 and 800 to 1200 vertices, respectively. Please refer to Appendix C for more details about the dataset description and statistics.

**Graph/SAT Model Backbones.** We implement multiple GNN backbones for the Representation Extractor in both graph and SAT models. For the graph models, we adopt two widely used backbones, GCN [31] and GraphSAGE [22], and two advanced backbones with stronger representational capacity, PGN [55] and GraphGPS [47]. For the SAT model, we implement NeuroSAT and a GCN architecture

Table 4: Performance on MVC. 'OBJ' refers to the average objective value, where lower is better for MVC. 'Optimal' represents the best-known solution obtained using Gurobi [21].

| Graph | Optimal | OptGNN | | OptGNN+ConRep4CO | | gain | GCNN | | GCNN+ConRep4CO | | gain |
|---|---|---|---|---|---|---|---|---|---|---|---|
| | | OBJ | $gap_{abs}$ | OBJ | $gap_{abs}$ | | OBJ | $gap_{abs}$ | OBJ | $gap_{abs}$ | |
| ER(50,100) | 54.62 | 55.87 | 1.25 | **54.70** | **0.08** | 93.60% | 55.34 | 0.72 | **55.17** | **0.55** | 23.61% |
| ER(100,200) | 122.79 | 126.04 | 3.25 | **124.37** | **1.58** | 51.40% | 128.29 | 5.50 | **126.75** | **3.96** | 28.00% |
| ER(400,500) | 417.42 | 420.51 | 3.09 | **419.31** | **1.89** | 38.83% | 443.43 | 26.01 | **436.77** | **19.35** | 25.61% |
| **avg. gain** | - | - | - | - | - | **61.27%** | - | - | - | - | **25.74%** |

Table 5: Performance on MIS, MC, and MDS. 'OBJ' refers to the average objective value, where higher is better for MIS and MC, and lower is better for MDS. 'Optimal' denotes the best-known solution, obtained using KAMIS [32] for MIS and Gurobi [21] for MC and MDS.

| Problem Type | Graph | Optimal | GFlowNet | | GFlowNet+ConRep4CO | | gain | avg. gain |
|---|---|---|---|---|---|---|---|---|
| | | | OBJ | $gap_{abs}$ | OBJ | $gap_{abs}$ | | |
| MIS↑ | RB(200,300) | 20.10 | 19.18 | 0.92 | 19.56 | **0.54** | 41.30% | |
| | RB(800,1200) | 43.15 | 37.48 | 5.67 | 38.79 | **4.36** | 23.10% | 32.20% |
| MC↑ | RB(200,300) | 19.05 | 16.24 | 2.81 | 17.47 | **1.58** | 43.77% | |
| | RB(800,1200) | 33.89 | 31.42 | 2.47 | 32.14 | **1.75** | 29.15% | 36.46% |
| MDS↓ | BA(200,300) | 27.89 | 28.61 | 0.72 | 28.19 | **0.30** | 58.33% | |
| | BA(800,1200) | 103.80 | 110.28 | 6.48 | 108.19 | **4.39** | 32.25% | 45.29% |

specifically tailored for SAT graphs. Moreover, we employ both LCG and VCG as SAT graph representations. Please refer to Appendix D for more details.

**Tasks.** The evaluation tasks can be divided into two parts: **1) Representation evaluation**, measured by the performance of graph models on the GDP-solving task, focusing on how the learned representations can accurately determine the solution for each specific problem type. The GDP-solving performance on more difficult instances and unseen GDP domains is used to assess both the in-domain (problem scale) and cross-domain generalizability of the learned representations. **2) Enhancement of CO problem-solving**, measured by the performance on four CO problems—MVC, MIS, MC, and MDS—when incorporating ConRep4CO into problem-specific neural solvers, as described in Sec. 3.3.4.

**Baselines.** For **1) representation evaluation**, to ensure a fair comparison, we establish baselines for the graph models trained by ConRep4CO by keeping the architectures identical while modifying only the training procedures. Each baseline graph model is trained independently on its corresponding dataset. For **2) enhancement of CO problem-solving**, OptGNN and GCNN from [72] are used as baseline neural solvers for MVC, while GFlowNet from [75] serves as the baseline neural solver for MIS, MC, and MDS. These models are trained using the methods described in their respective papers and compared to those incorporating ConRep4CO as outlined in Sec. 3.3.4.

## 4.2 REPRESENTATION EVALUATION

### 4.2.1 REPRESENTATION QUALITY EVALUATION

We evaluate the quality of the learned representations by comparing the accuracy of graph models in solving seven GDPs. The baseline model is referred to as **Graph Model**, which is trained independently on its corresponding dataset. Our proposed approach, denoted **Graph Model+ConRep4CO**, initializes the model parameters with a pre-trained checkpoint from ConRep4CO trained on the seven GDP datasets and is then fine-tuned individually. Table 1 presents the results for six combinations of SAT and graph backbones, evaluating the performance of both models trained and tested on datasets with identical distributions, including the easy and medium difficulty datasets. The proposed approach consistently outperforms the baseline model across most GDP tasks, at both difficulty levels, and for all six backbone combinations. These findings indicate that integrating ConRep4CO substantially enhances the quality of the learned representations, enabling more effective capture of the underlying features and characteristics of GDPs. Consequently, the enhanced representations lead to improved accuracy in solving GDPs. Notably, when employing the GraphSAGE backbone, our approach demonstrates a particularly significant performance improvement over the baseline.

Table 6: Ablation study on cross-domain information transfer. The table presents GDP-solving accuracy (%) with confidence intervals ($\alpha = 0.05$). 'Graph Model+ConRep4CO+Single Domain' refers to the ablated method by disabling cross-domain information transfer.

| Difficulty | Model | $k$-Clique | $k$-Domset | $k$-Vercov | $k$-Color | $k$-Indset | Matching | Automorph | Overall |
|---|---|---|---|---|---|---|---|---|---|
| Easy | Graph Model+ConRep4CO+Single Domain | 78.4±0.4 | 61.7±0.1 | 67.1±0.1 | 89.9±0.2 | 65.6±0.4 | 71.5±0.2 | 65.3±0.2 | 71.4 |
| | Graph Model+ConRep4CO | **79.3±0.3** | **62.0±0.1** | **67.3±0.2** | **90.2±0.1** | **67.5±0.5** | **71.7±0.3** | **65.4±0.3** | **71.9** |
| Medium | Graph Model+ConRep4CO+Single Domain | 70.9±0.3 | 64.0±0.1 | 62.9±0.1 | 81.0±0.3 | 59.9±0.4 | 72.5±0.2 | 64.3±0.4 | 67.9 |
| | Graph Model+ConRep4CO | **71.3±0.5** | **64.6±0.2** | **63.3±0.3** | **82.2±0.2** | **64.0±0.1** | **72.8±0.4** | **65.7±0.4** | **69.1** |

### 4.2.2 IN-DOMAIN GENERALIZABILITY EVALUATION

To assess the in-domain generalization capabilities of the learned representations, particularly in relation to problem scale, we evaluate their performance on previously unseen hard GDP datasets, which consist of problem instances with increased scale and complexity. Table 2 presents the results for six combinations of SAT and graph backbones, with graph models trained on the easy and medium datasets and tested on the hard datasets. The results clearly demonstrate that the representations learned by ConRep4CO show improved performance across most GDP tasks, both difficulty levels, and all six backbone combinations. This indicates that ConRep4CO also improves generalizability to more challenging and complex problem instances that were previously unseen. The improvement likely stems from the information transfer between various GDPs during pre-training, allowing the model to learn shared, more general, and high-level representations. This generality is applicable to problem scale, further reinforcing the robustness of the learned representations.

### 4.2.3 CROSS-DOMAIN GENERALIZABILITY EVALUATION

To further evaluate cross-domain generalizability, particularly the ability to generalize to unseen problem domains during pre-training, we select 6 out of 7 GDPs as pre-training domains, with the remaining GDP serving as the generalizing domain. After pre-training, we define a new graph model for the generalizing domain, align it with the pre-trained SAT model from the pre-training domains, and fine-tune it on 20,000 instances from the generalizing domain. The graph model for the generalizing domain is referred to as **Graph Model-Unseen**. We conduct experiments on all combinations of pre-training domains and report the GDP-solving accuracy for the generalizing domain in Table 3. Note that each number in the second row corresponds to a complete independent experiment, where the dataset indicated in the header is the generalizing domain, and the remaining six GDPs serve as pre-training domains. The results demonstrate that ConRep4CO enables the graph model to learn better representations for an unseen GDP domain with minimal data, outperforming the baseline. This indicates that ConRep4CO not only enhances representation quality for specific tasks but also improves representation learning for problem domains that are completely unseen. The improvement is likely due to the unified SAT model capturing shared features among pre-training domains, allowing it to learn a general GDP representation applicable to unseen GDPs. Aligning with the SAT model helps the graph model learn new GDP representations more effectively and efficiently.

The results also reveal that it is not necessary to include all GDP domains during pre-training when scaling to a large number of GDPs. Aligning with the pre-trained SAT model allows for effective transfer to new GDPs with minimal data, showing the practical applicability of ConRep4CO.

### 4.3 ENHANCEMENT OF CO PROBLEM-SOLVING

We incorporate ConRep4CO into the training of neural solvers for four CO problems to evaluate how it enhances general CO problem-solving. The incorporation process is described in Sec. 3.3.4, denoted as **+ConRep4CO**. The pre-training datasets are the $k$-Vercov dataset for MVC, the $k$-Indset for MIS, the $k$-Clique for MC, and the $k$-Domset for MDS, all with the easy difficulty level. Table 4 presents the results for MVC, while Table 5 shows the results for MIS, MC, and MDS. 'gap$_{\text{abs}}$' refers to the absolute gap between the output objective and the optimal value, and 'gain' represents the improvement from incorporating ConRep4CO over the baseline, which is calculated as $(\text{gap}_{\text{abs, baseline}} - \text{gap}_{\text{abs, ours}})/\text{gap}_{\text{abs, baseline}} \times 100\%$. 'avg. gain' is the average gain for each problem across all problem scales. Incorporating ConRep4CO yields average gains of 61.27% (OptGNN), 32.20%, 36.46%, and 45.29% for MVC, MIS, MC, and MDS, respectively. We observe that the gain tends to be smaller for larger-scale problems, likely due to the increased difficulty and the growing discrepancy between the scales of the pre-training datasets and the evaluated problem. However, across all problem scales, we consistently see substantial performance enhancements. These results

Table 7: The effect of the number of pre-training domains on downstream performance. Values in parentheses represent the gain over baseline.

| Problem | Graph | Optimal | Baseline | Ours-1 | Ours-2 | Ours-3 | Ours-4 | Ours-5 | Ours-6 | Ours-7 |
|---|---|---|---|---|---|---|---|---|---|---|
| MVC | ER(50,100) | 54.62 | 55.87 | 55.75 (9.60%) | 55.39 (38.40%) | 55.07 (64.00%) | 54.76 (88.80%) | 54.74 (90.40%) | 54.68 (95.20%) | 54.70 (93.60%) |
| MVC | ER(100,200) | 122.79 | 126.04 | 125.86 (5.54%) | 125.10 (28.92%) | 124.89 (35.38%) | 124.51 (47.08%) | 124.44 (49.23%) | 124.39 (50.77%) | 124.37 (51.40%) |
| MVC | ER(400,500) | 417.42 | 420.51 | 420.40 (3.56%) | 419.86 (21.04%) | 419.63 (28.48%) | 419.49 (33.01%) | 419.39 (36.25%) | 419.33 (38.19%) | 419.31 (38.83%) |
| MIS | RB(200,300) | 20.10 | 19.18 | 19.22 (4.35%) | 19.47 (31.52%) | 19.52 (36.96%) | 19.55 (40.22%) | 19.53 (38.04%) | 19.57 (42.39%) | 19.56 (41.30%) |
| MIS | RB(800,1200) | 43.15 | 37.48 | 37.62 (2.47%) | 38.47 (17.46%) | 38.66 (20.81%) | 38.74 (22.22%) | 38.79 (23.10%) | 38.77 (22.75%) | 38.79 (23.10%) |
| MC | RB(200,300) | 19.05 | 16.24 | 16.41 (6.05%) | 16.90 (23.49%) | 17.23 (35.23%) | 17.36 (39.86%) | 17.44 (42.70%) | 17.47 (43.77%) | 17.47 (43.77%) |
| MC | RB(800,1200) | 33.89 | 31.42 | 31.49 (2.83%) | 31.75 (13.36%) | 31.90 (19.43%) | 32.07 (26.31%) | 32.11 (27.94%) | 32.12 (28.34%) | 32.14 (29.15%) |
| MDS | RB(200,300) | 27.89 | 28.61 | 28.56 (6.94%) | 28.42 (26.39%) | 28.28 (45.83%) | 28.24 (51.39%) | 28.18 (59.72%) | 28.18 (59.72%) | 28.19 (58.33%) |
| MDS | RB(800,1200) | 103.80 | 110.28 | 110.04 (3.70%) | 109.12 (17.90%) | 108.68 (24.69%) | 108.34 (29.94%) | 108.22 (31.79%) | 108.17 (32.56%) | 108.19 (32.25%) |

demonstrate that ConRep4CO enhances general CO problem-solving beyond GDPs by effectively integrating with problem-specific neural solvers, highlighting its strong application potential.

## 4.4 ANALYSIS ON CROSS-DOMAIN INFORMATION TRANSFER

A central component of our framework is the facilitation of information transfer across different problem domains. To evaluate the effectiveness of this mechanism, we conduct an ablation study by disabling the cross-domain information transfer. Specifically, we train each graph model independently with its own SAT model, without leveraging cross-domain information. We then compare this ablated approach with our original method, as shown in Table 6. The results indicate that the ablated approach yields inferior performance, thereby highlighting the importance and effectiveness of the cross-domain information transfer in enhancing representation learning.

## 4.5 FURTHER STUDY ON MULTIPLE DOMAIN INFORMATION TRANSFER

This section investigates the impact of multi-domain information transfer on model performance. We address two key questions: 1) **What is the effect of transferring information from multiple domains to downstream CO tasks?** 2) **How does this effect scale with the number of source domains?** This analysis aims to provide a comprehensive evaluation of our proposed cross-domain transfer mechanism. We design seven pre-training configurations, utilizing data from 1 to 7 distinct problem domains, denoted as **Ours-1** to **Ours-7**. Each pre-trained SAT model is subsequently used to enhance problem-specific neural solvers via contrastive learning. We employ OptGNN [72] as the baseline solver for MVC and GFlowNet [75] for MIS, MC, and MDS. The alignment between the pre-trained SAT models and the solvers is performed on 5,000 easy-level instances, with an additional 1,000 instances for validation. The subsequent training procedures for the solvers remain consistent with their original implementations [72; 75]. The results are presented in Table 7. The **Ours-1** configuration, which does not leverage cross-domain information transfer, shows a modest performance gain of approximately 5%. This result primarily reflects the benefit of the SAT transformation itself and suggests that pre-training on a single domain provides limited representational enhancement. As the number of pre-training domains increases, the performance gain grows in a sublinear fashion. The improvement is sharpest when expanding from one to two domains, after which the rate of gain decelerates, converging at around five or six domains. This saturation effect is likely attributable to diminishing returns in novel information.

## 5 CONCLUSION AND OUTLOOK

We have introduced ConRep4CO, a novel contrastive paradigm designed to promote learning representation for CO problems across problem types. The results indicate that it not only improves the quality and generalizability of the learned representations but also significantly enhances general CO problem-solving, highlighting the potential of ConRep4CO as a unified pre-training paradigm for CO research. Future work will focus on exploring unsupervised approaches to reduce reliance on labeled data, thereby increasing applicability in data-sparse scenarios. Additionally, we will investigate other potential forms to replace the currently used SAT form to perform contrastive learning, aiming for broader application to more general CO problems, such as mixed-integer linear programs (MILP).

ETHICS STATEMENT

This research follows the ICLR Code of Ethics and ensures that no ethical concerns are overlooked. The study does not involve human subjects or sensitive personal data. All datasets used are either synthetic or publicly available, and the methods for data generation and transformation are transparent and reproducible. We have tried to minimize bias in our experiments and ensure that the results do not perpetuate unfair outcomes. Additionally, the authors declare no conflicts of interest or financial sponsorships that could have influenced the research. Our work is committed to contributing positively to the field of combinatorial optimization.

REPRODUCIBILITY STATEMENT

We prioritize reproducibility and have taken steps to ensure that our experiments can be independently replicated. Detailed descriptions of the datasets are provided in Appendix C. Additionally, the detailed network architectures are outlined in Appendix D.

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

# A    DISCUSSION WITH RELATED WORK

In this section, we discuss the key differences of our ConRep4CO with some prior works [12; 7] proposing general representation learning methods for CO.

## A.1    COMPARISON WITH [12]

**GOAL** [12] aims to develop a generalist model that uses a single backbone to represent multiple CO problems. The model employs a multi-type Transformer architecture and attention blocks, which suggest that different CO problems activate distinct portions of the backbone parameters. Consequently, GOAL accommodates different problems by utilizing different parts of the shared parameters, rather than exploiting the commonalities between different problems. Furthermore, GOAL's performance is slightly inferior to problem-specific baselines, indicating that the learned representations are not enhanced for each problem.

In contrast, ConRep4CO focuses on capturing the shared underlying structure across multiple CO problems, seeking a higher-level, abstract representation that encapsulates the essence common to different problem domains. By using SAT as a unified intermediary, ConRep4CO facilitates knowledge transfer and mutual enhancement among problems, enabling insights gained from one problem to improve the performance on others. Therefore, ConRep4CO outperforms baselines trained on individual problems.

Overall, ConRep4CO differs from GOAL in its emphasis on leveraging the shared structure of multiple CO problems to improve the learned representation for individual problems, rather than simply accommodating problem-specific variations within a single model.

## A.2    COMPARISON WITH [7]

The MILP multi-task framework [7] focuses on learning shared representations across tasks within a single problem domain. It seeks to exploit commonalities among different tasks within the MILP domain. One could draw an analogy between the front-end network architecture in the framework and the representation extractor in our graph model, with the task-specific layers serving as different output modules. The MILP multi-task framework emphasizes improving these output modules with a single representation extractor, while ConRep4CO prioritizes enhancing the representation extractor by knowledge transfer and mutual enhancement across diverse problem domains. By doing so, ConRep4CO learns a higher-level, abstract representation that spans various CO problems, whereas the MILP framework primarily concentrates on task-specific output layers for a single problem domain.

To the best of our knowledge, ConRep4CO is the first framework to leverage representations across different problem domains to improve representations for individual problem domains.

# B    DISCUSSION ON APPLICABILITY OF CONREP4CO

During the pre-training phase, our framework is designed to handle any NP problem. As long as a problem can be reduced to SAT, it can be trained within our framework. We would like to clarify that all NP problems can be reduced to NPC problems, which in turn can be transformed into SAT problems. This theoretical foundation ensures that our framework is applicable to all NP decision problems, regardless of their specific structure or variable types. In this sense, our approach is problem-agnostic, enabling effective training on a wide range of NP decision problems.

Furthermore, our approach can facilitate vast NP-hard CO problems by pre-training on their decision versions, which are typically NPC. Most CO problems can be converted into their decision versions by adding a target value. In our approach, we can pre-train the models on the decision versions of these CO problems to learn effective representations. These representations can be applied not only to the decision versions but also to the optimization versions of these problems (original problems).

Overall, ConRep4CO can serve as a unified pre-training paradigm for a broad range of CO problems.

Table 8: Details of generated GDP datasets.

| Dataset | Description | Parameters | Notes |
|---|---|---|---|
| $k$-Clique | The $k$-Clique dataset consists of graph instances of the $k$-Clique problem, which involves determining whether a given graph contains a clique of size $k$. A clique is a subset of vertices in which every pair of vertices is connected by an edge. The goal is to identify whether such a fully connected subset of $k$ vertices exists within the graph. Instances are built on randomly generated Erdős-Rényi graphs. Parameters include number of vertices $v$, edge probabilities $p$, and clique size $k$. | General: $p = \binom{v}{k}^{-1/\binom{v}{2}}$,
Easy dataset: $v \sim$ Uniform(5, 15), $k \sim$ Uniform(3, 4),
Medium dataset: $v \sim$ Uniform(15, 20), $k \sim$ Uniform(3, 5),
Hard dataset: $v \sim$ Uniform(20, 25), $k \sim$ Uniform(4, 6). | The parameter $p$ is selected based on [5], ensuring that the expected number of $k$-cliques in the generated graph is equal to 1. |
| $k$-Domset | The $k$-Domset dataset consists of graph instances of the $k$-Dominating Set problem, which involves determining whether a given graph contains a dominating set of size $k$. A dominating set is a subset of vertices such that every vertex in the graph is either in the subset or adjacent to at least one vertex in the subset. The goal is to identify whether such a subset of $k$ vertices exists that can 'dominate' the entire graph, ensuring that all other vertices are either in the subset or connected to it. Instances are built on randomly generated Erdős-Rényi graphs. Parameters include number of vertices $v$, edge probabilities $p$, and dominating set size $k$. | General: $p = 1 - \left(1 - \binom{v}{k}^{-1/(v-k)}\right)^{1/k}$,
Easy dataset: $v \sim$ Uniform(5, 15), $k \sim$ Uniform(2, 3),
Medium dataset: $v \sim$ Uniform(15, 20), $k \sim$ Uniform(3, 5),
Hard dataset: $v \sim$ Uniform(20, 25), $k \sim$ Uniform(4, 6). | The parameter $p$ is selected based on [66], ensuring that the expected number of $k$-dominating sets in the generated graph is equal to 1. |
| $k$-Vercov | The $k$-Vercov dataset consists of graph instances of the $k$-Vertex Cover problem, which involves determining whether a given graph contains a vertex cover of size $k$. A vertex cover is a subset of vertices such that every edge in the graph is incident to at least one vertex in the subset. The goal is to identify whether a subset of $k$ vertices exists that can 'cover' all the edges in the graph, ensuring that each edge is connected to at least one vertex in the subset. Instances are built on randomly generated Erdős-Rényi graphs. Parameters include number of vertices $v$, edge probabilities $p$, and vertex set size $k$. | General: $p = \binom{v}{k}^{-1/\binom{v}{2}}$,
Easy dataset: $v \sim$ Uniform(5, 15), $k \sim$ Uniform(3, 5),
Medium dataset: $v \sim$ Uniform(10, 20), $k \sim$ Uniform(6, 8),
Hard dataset: $v \sim$ Uniform(15, 25), $k \sim$ Uniform(9, 10). | The parameter $p$ is selected based on the relationship between $k$-Clique and $k$-Vercov, ensuring that the expected size of the minimum vertex cover in the generated graph is $k$. |
| $k$-Color | The $k$-Color dataset consists of graph instances of the $k$-Coloring problem, which involves determining whether a given graph can be colored with $k$ colors such that no two adjacent vertices share the same color. A valid coloring assigns one of $k$ different colors to each vertex, ensuring that vertices connected by an edge have different colors. The goal is to identify whether such a coloring scheme exists for the graph using at most $k$ colors. Instances are built on randomly generated Erdős-Rényi graphs. Parameters include number of vertices $v$, edge probabilities $p$, and number of colors $k$. | General: $p = \binom{v}{k}^{-1/\binom{v}{2}}$,
Easy dataset: $v \sim$ Uniform(5, 15), $k \sim$ Uniform(3, 4),
Medium dataset: $v \sim$ Uniform(15, 20), $k \sim$ Uniform(3, 5),
Hard dataset: $v \sim$ Uniform(20, 25), $k \sim$ Uniform(4, 6). | The parameter $p$ is selected based on the relationship between $k$-Clique and $k$-Color, ensuring that the expected minimum number of colors for the generated graph is $k$. |
| $k$-Indset | The $k$-Indset dataset consists of graph instances of the $k$-Independent Set problem, which involves determining whether a given graph contains an independent set of size $k$. An independent set is a subset of vertices in which no two vertices are adjacent, meaning there are no edges connecting any pair of vertices in the subset. The goal is to identify whether such a subset of $k$ vertices exists within the graph, ensuring that the selected vertices are mutually non-adjacent. Instances are built on randomly generated Erdős-Rényi graphs. Parameters include number of vertices $v$, edge probabilities $p$, and independent set size $k$. | General: $p = 1 - \binom{v}{k}^{-1/\binom{v}{2}}$,
Easy dataset: $v \sim$ Uniform(5, 15), $k \sim$ Uniform(3, 4),
Medium dataset: $v \sim$ Uniform(15, 20), $k \sim$ Uniform(3, 5),
Hard dataset: $v \sim$ Uniform(20, 25), $k \sim$ Uniform(4, 6). | The parameter $p$ is selected based on the relationship between $k$-Clique and $k$-Indset, ensuring that the expected number of $k$-independent sets in the generated graph is equal to 1. |
| Matching | The Matching dataset consists of graph instances of the Perfect Matching problem, which involves determining whether a given graph contains a perfect matching. A perfect matching is a subset of edges in which every vertex in the graph is incident to exactly one edge in the subset. In other words, the graph's vertices can be paired off so that no vertex is left unpaired and no two edges share a vertex. The goal is to identify whether such a perfect matching exists within the graph, ensuring that all vertices are perfectly matched. Instances are built on randomly generated Erdős-Rényi graphs. Parameters include number of vertices $v$ and edge probabilities $p$. | General: $p = \ln(v)/v$,
Easy dataset: $v \sim$ Uniform(6, 16), should be an even number,
Medium dataset: $v \sim$ Uniform(16, 24), should be an even number,
Hard dataset: $v \sim$ Uniform(24, 30), should be an even number. | The selected parameter $p$ is a sharp threshold for graph connectivity based on [14], ensuring that the generated graph is neither too dense nor too sparse. |
| Automorph | The Automorph dataset consists of graph instances of the Graph Automorphism problem, which involves determining whether a given graph has a non-trivial automorphism. An automorphism is a mapping of the graph's vertices to itself such that the structure of the graph is preserved, meaning that the adjacency relationships between vertices remain unchanged. The goal is to identify whether there exists a way to rearrange the vertices of the graph such that it appears identical to its original form. Instances are built on randomly generated Erdős-Rényi graphs. Parameters include number of vertices $v$ and edge probabilities $p$. | General: $p = \ln(v)/v$,
Easy dataset: $v \sim$ Uniform(4, 8),
Medium dataset: $v \sim$ Uniform(8, 10),
Hard dataset: $v \sim$ Uniform(10, 12). | The selected parameter $p$ is a sharp threshold for graph connectivity based on [14], ensuring that the generated graph is neither too dense nor too sparse. |

## C   MORE DETAILS ON DATASETS

In this section, we provide more details on the utilized datasets in our main paper, including the parameters of GDP instances and the statistics of SAT instances.

### C.1   GDP INSTANCES

To ensure the generation of high-quality GDP instances that accurately capture the inherent characteristics of each problem, we carefully select the graph distributions and parameters used for instance generation. Some parameters refer to [40]. Table 8 provides a detailed overview of the specific GDP datasets employed in the main paper.

Note that six of the seven GDPs are NP-complete, while the Perfect Matching problem is a P problem.

Table 9: SAT dataset statistics. # Variables refers to average number of variables, # Clauses denoted average number of clauses, Mod. (LCG) represents average modularity of LCG graphs, and Mod. (VCG) represents average modularity of VCG graphs.

| Dataset | Easy | | | | Medium | | | | Hard | | | |
|---|---|---|---|---|---|---|---|---|---|---|---|---|
| | # Variables | # Clauses | Mod. (LCG) | Mod. (VCG) | # Variables | # Clauses | Mod. (LCG) | Mod. (VCG) | # Variables | # Clauses | Mod. (LCG) | Mod. (VCG) |
| $k$-Clique | 35.69 | 613.25 | 0.49 | 0.46 | 70.86 | 2298.03 | 0.49 | 0.48 | 114.49 | 5670.10 | 0.50 | 0.49 |
| $k$-Domset | 40.73 | 345.75 | 0.53 | 0.47 | 89.70 | 1708.06 | 0.51 | 0.49 | 137.32 | 4025.85 | 0.51 | 0.49 |
| $k$-Vercov | 46.33 | 498.06 | 0.52 | 0.48 | 108.19 | 2681.55 | 0.51 | 0.49 | 192.57 | 8409.32 | 0.51 | 0.50 |
| $k$-Color | 33.91 | 112.64 | 0.69 | 0.65 | 69.92 | 321.25 | 0.71 | 0.68 | 112.16 | 719.32 | 0.69 | 0.66 |
| $k$-Indset | 38.38 | 702.92 | 0.49 | 0.46 | 72.55 | 2388.22 | 0.49 | 0.48 | 113.12 | 5549.79 | 0.50 | 0.49 |
| Matching | 27.48 | 95.03 | 0.69 | 0.59 | 30.92 | 107.67 | 0.70 | 0.61 | 45.48 | 169.49 | 0.72 | 0.64 |
| Automorph | 56.76 | 943.54 | 0.51 | 0.47 | 82.74 | 1856.26 | 0.51 | 0.48 | 121.56 | 3612.56 | 0.51 | 0.49 |

## C.2 SAT INSTANCES

After generating the seven GDP datasets, the corresponding seven SAT datasets are generated by transforming the GDP datasets, utilizing the python toolkit CNFGen [33]. We also compute the statistics of those SAT datasets to provide comprehensive information on datasets. The dataset statistics are shown in Table 9.

Moreover, to evaluate the effectiveness of the learned representations on unseen SAT instances, we synthetically generate four more SAT datasets, including two random problems and two pseudo-industrial problems. Specifically, for random problems, we generate the SR dataset with the SR generator in NeuroSAT [50], and the 3-SAT dataset with the 3-SAT generator in CNFGen [33]. For pseudo-industrial problems, we generate the CA dataset via the Community Attachment model [18], and the PS dataset by the Popularity-Similarity model [19]. The generation process of the four datasets follows [40], where the dataset descriptions and statistics can also be found.

The ground truth of satisfiability and satisfying assignments are calculated by calling the state-of-the-art modern SAT solver CaDiCaL [15], and the truth labels for unsat core variables are generated by invoking the proof checker DRAT-trim [65].

# D DETAILS ON MODEL ARCHITECTURE

## D.1 BASIC ARCHITECTURE IMPLEMENTATION

**Graph Model.** Each graph model is designed to address a specific type of GDP, and all models maintain a consistent architecture. To illustrate this, we focus on problem $\mathcal{P}_n$ and its corresponding graph model $\mathbb{M}_n$. The graph model $\mathbb{M}_n$ takes graphs in the set $\mathbf{G}_n$ as input and processes them through the Representation Extractor. The input graph primarily consists of edge information, which is often a critical aspect of GDPs. For the initial vertex features, we introduce a $d$-dimensional embedding for all vertices, represented as $\mathbf{h}_n^{(0)}$.

For the Representation Extractor, we adopt the vanilla Graph Convolutional Network (GCN) [31], which is widely used as a backbone for node embeddings in graph-based tasks. Assume there are $k$ layers, the embedding extraction at the $i$-th layer of the network is expressed as:

$$\mathbf{H}_n^{(i)} = \text{ReLU}(\tilde{\mathbf{D}}^{-\frac{1}{2}}\tilde{\mathbf{A}}\tilde{\mathbf{D}}^{-\frac{1}{2}}\mathbf{H}_n^{(i-1)}\mathbf{W}_n^{(i-1)}), \; i = 1, 2, \ldots, k, \tag{3}$$

where $\mathbf{H}$ denotes the node embedding matrix, with each row corresponding to a node embedding. The matrix $\tilde{\mathbf{A}} = \mathbf{A} + \mathbf{I}$ is the adjacency matrix augmented with self-loops through the identity matrix $\mathbf{I}$. $\tilde{\mathbf{D}}_{ii} = \sum_j \tilde{\mathbf{A}}_{ij}$ is the degree matrix, and $\mathbf{W}$ is the learnable weight matrix. Following the extraction of node features, we apply average pooling to the node embedding matrix $\mathbf{H}_n^{(k)}$ to aggregate the node-level information into a single representation for the entire graph instance, denoted as $\mathbf{r}_n$. This aggregation is computed as follows:

$$\mathbf{r}_n = \frac{\sum_{v \in \mathcal{V}} \mathbf{h}_{n,v}^{(k)}}{|\mathcal{V}|}, \tag{4}$$

where $\mathcal{V}$ represents the set of vertices in the input graph, $|\mathcal{V}|$ denotes the total number of vertices, and $\mathbf{h}_{n,v}^{(k)}$ is the extracted embedding for node $v$. $\mathbf{r}_n$ serves as the instance-level feature representation, and is subsequently fed into the Output Module, which is implemented as an MLP to produce the final decision for the instance.

**SAT Model.** Apart from the graph models, the SAT model $\mathbb{M}_{sat}$ processes the constructed SAT graphs via its own Representation Extractor. For illustration, we consider the LCG representation. For the initial node features, we define two distinct $d$-dimensional embeddings: $\mathbf{h}_l^{(0)}$ for all literal nodes and $\mathbf{h}_c^{(0)}$ for all clause nodes.

The architecture of the Representation Extractor is inspired by NeuroSAT [50]. For notational clarity, we assume that the extractor consists of $k$ layers, with both literal and clause node embeddings being iteratively aggregated and updated at each layer. At the $i$-th layer, the updates for the literal and clause node embeddings are formulated as follows:

$$\mathbf{h}_l^{(i)} = \text{LayerNormLSTM} \left( \underset{c \in \mathcal{N}(l)}{\text{SUM}} \left( \text{MLP} \left( \mathbf{h}_c^{i-1} \right) \right), \mathbf{h}_l^{(i-1)}, \mathbf{h}_{\neg l}^{(i-1)} \right), \tag{5}$$

$$\mathbf{h}_c^{(i)} = \text{LayerNormLSTM} \left( \underset{l \in \mathcal{N}(c)}{\text{SUM}} \left( \text{MLP} \left( \mathbf{h}_l^{i-1} \right) \right), \mathbf{h}_c^{(i-1)} \right), \tag{6}$$

where $l$ and $c$ represent an arbitrary literal node and clause node, respectively, $\mathcal{N}(\cdot)$ refers to the set of neighboring nodes. The summation operator (SUM) serves as the aggregation function, while LayerNormLSTM [1] is employed as the update function.

Similar to the graph models, the instance-level representation $\mathbf{r}_{sat}$ derives by averaging the literal node embeddings after the $k$-th layer. The instance-level representation, along with the literal-level embeddings, is passed to the Output Module, which is also implemented as an MLP, to generate the final task-specific decisions or predictions.

## D.2 INITIAL VERTEX FEATURES

As illustrated in the main paper, the input graphs primarily provide edge information instead of vertex features. Therefore, we should devise initial vertex features for the models. In this section, we introduce the definition of initial vertex features for the graph and SAT models.

**Graph Model Vertex Feature.** We begin by generating a normalized, learnable $d$-dimensional vector, which serves as the initial embedding shared across all vertices. For GDP datasets that do not require additional problem-specific information, such as Matching and Automorph, this initial embedding is directly used as the vertex feature for all vertices. In contrast, for GDP datasets where the parameter $k$ plays a critical role in defining the instance characteristics, such as $k$-Clique and $k$-Vercov, we first embed $k$ into a $d$-dimensional vector. The initial vertex embedding is then fused with the $k$ embedding through an MLP to generate the final initial vertex features.

**SAT Model Vertex Feature.** For the SAT model, we generate initial vertex features based on the type of SAT graph representation, whether it is a Literal-Clause Graph (LCG) or a Variable-Clause Graph (VCG). In the case of the LCG graph, we initialize a normalized, learnable $d$-dimensional vector for all literal nodes and a separate normalized, learnable $d$-dimensional vector for all clause nodes. Similarly, for the VCG graph, we generate a normalized, learnable $d$-dimensional vector for all variable nodes and another for all clause nodes.

## D.3 MORE BACKBONES

To demonstrate that the performance improvement brought about by our ConRep4CO is consistent, and independent with specialized model architectures, we conduct experiments on more backbones.

**Graph Model Backbone.** For the graph model, we employ an additional mainstream network architecture for node embedding, GraphSAGE [22], which is widely recognized for its ability to generate inductive representations of graph nodes by aggregating information from a node's local neighborhood. The update rule for the $i$-th layer of GraphSAGE is defined as follows:

$$\mathbf{n}_u^{(i)} = \text{AGG} \left( \text{ReLU} \left( \mathbf{Q}^{(i)} \mathbf{h}_v^{(i)} + \mathbf{q}^{(i)} \mid v \in N(u) \right) \right), \tag{7}$$

$$\mathbf{h}_u^{(i+1)} = \text{ReLU} \left( \mathbf{W}^{(i)} \text{CONCAT} \left( \mathbf{h}_u^{(i)}, \mathbf{n}_u^{(i)} \right) \right), \tag{8}$$

where $\mathbf{h}_u$ denotes the embedding for vertex $u$, $N(u)$ refers to the neighbors of vertex $u$, $\mathbf{Q}, \mathbf{q}, \mathbf{W}$ are trainable parameters, and AGG is the aggregation function. In our implementation, AGG is defined as the mean function, which computes the element-wise average of the neighbor embeddings.

In addition, we implement two more advanced GNN backbones for our graph models, PGN [55] and GraphGPS [47]. Please refer to the corresponding papers for details on the architectures of these two backbones.

**SAT Model Backbone.** For the SAT model, we incorporate a GCN architecture specifically tailored for SAT graphs as an additional backbone. The node updates at the $i$-th layer are defined as follows:

$$\mathbf{h}_l^{(i)} = \text{MLP}\left(\underset{c \in \mathcal{N}(l)}{\text{SUM}}\left(\text{MLP}\left(\mathbf{h}_c^{i-1}\right)\right), \mathbf{h}_l^{(i-1)}, \mathbf{h}_{\neg l}^{(i-1)}\right), \tag{9}$$

$$\mathbf{h}_c^{(i)} = \text{MLP}\left(\underset{l \in \mathcal{N}(c)}{\text{SUM}}\left(\text{MLP}\left(\mathbf{h}_l^{i-1}\right)\right), \mathbf{h}_c^{(i-1)}\right), \tag{10}$$

where $l$ and $c$ represent an arbitrary literal node and clause node, respectively. The aggregation of neighboring node information is performed using the summation operator (SUM), which serves as the aggregation function. The updates for both literal and clause nodes are computed using an MLP.

Furthermore, we extend the backbone to VCG graph modeling, where all literal nodes are replaced by variable nodes, and each literal and its negation are merged into a single variable node. The node updates at the $i$-th layer of the VGC-based GCN are formulated as:

$$\mathbf{h}_v^{(i)} = \text{MLP}\left(\underset{c \in \mathcal{N}(v)}{\text{SUM}}\left(\text{MLP}\left(\mathbf{h}_c^{i-1}\right)\right), \mathbf{h}_v^{(i-1)}\right), \tag{11}$$

$$\mathbf{h}_c^{(i)} = \text{MLP}\left(\underset{v \in \mathcal{N}(c)}{\text{SUM}}\left(\text{MLP}\left(\mathbf{h}_v^{i-1}\right)\right), \mathbf{h}_c^{(i-1)}\right), \tag{12}$$

where $v$ and $c$ represent an arbitrary variable node and clause node, respectively.

### D.4 CASE STUDY ON MODEL OUTUT

In this section, we illustrate the model outputs for specific GDP and corresponding SAT problems for better understanding.

In the context of GDP, the model's output is typically binary, represented as 0 or 1, at the instance level. For instance, in the case of the $k$-Clique problem, the input consists of a graph, and the output indicates whether the graph contains a clique of size $k$. Specifically, if a $k$-Clique is present, the output is 1; otherwise, it is 0.

Similarly, for the corresponding SAT problem, the output denotes the satisfiability of the formula. If the formula is satisfiable, the output is 1; if not, it is 0. The satisfiability result is directly linked to the solution of the original GDP problem. For example, a satisfiable formula indicates the existence of a $k$-Clique in the original graph.

However, the framework is not restricted to this specific task alone. By making appropriate modifications to the architecture of the output module, the models can be adapted to solve other related tasks, including both SAT-based and GDP-based tasks.

## E TRAINING DETAILS

### E.1 TRAINING PARAMETERS

For reproducibility, we present some important parameters used for training in Table 10. More details can be found in our source code, which will be released once the paper is accepted.

### E.2 COMPUTATIONAL COST

All training and inference tasks were conducted on a single NVIDIA H100 GPU with 80GB of memory.

Table 10: Parameters used for training.

| Parameter | Value | Description |
|---|---|---|
| lr | 1e-04 | Learning rate. |
| lr_step_size | 50 | Learning rate step size. |
| lr_factor | 0.5 | Learning rate factor. |
| lr_patience | 10 | Learning rate patience. |
| clip_norm | 1.0 | Clipping norm. |
| weight_decay | 1e-08 | L2 regularzation weight. |
| sat_model_gnn_layer | 32 | Iteration number of GNN layers in SAT model. |
| graph_model_gnn_layer | 12 | Iteration number of GNN layers in graph model. |
| mlp_layer | 2 | Number of Linear layers in an MLP. |
| $\tau$ | 0.1 (easy) / 0.5 (medium) | Temperature scalar in the contrastive loss. |
| $\beta$ | 0.5~1.0 | Weight of the decision loss during training. |

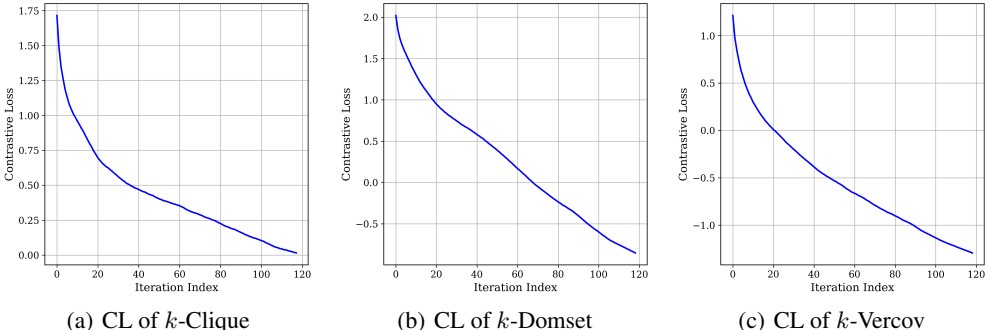

(a) CL of $k$-Clique        (b) CL of $k$-Domset        (c) CL of $k$-Vercov

Figure 4: Contrastive loss w.r.t. training iterations across various datasets. CL denotes the contrastive loss of the training process.

The pre-training process for the SAT model and the graph models with ConRep4CO totally takes approximately 40 hours, with convergence typically occurring around the 20th epoch. Each epoch requires roughly 2 hours. Following the pre-training phase, fine-tuning takes an additional 5 to 6 hours for each model to achieve optimal performance. In comparison, training the baseline SAT model takes about 45 hours, with convergence reached by the 30th epoch, and each epoch requiring approximately 1.5 hours. Notably, pre-training with ConRep4CO demonstrates a faster convergence rate, leading to a shorter training time. Moreover, training the baseline graph model independently each requires around 15 hours, with convergence occurring around the 60th epoch, and each epoch taking between 12 to 18 minutes.

Overall, the computational cost of training with ConRep4CO is comparable to that of the conventional training approach, with no significant increase in computational burden.

## F   FURTHER STUDIES

### F.1   FURTHER STUDY ON CONTRASTIVE LOSS

We revise the negative sampling strategy within our contrastive learning framework to mitigate the issue of false negative samples. Specifically, within each training batch, unsatisfiable instances are selected as negative samples for satisfiable instances, and conversely, satisfiable instances are chosen as negative samples for unsatisfiable instances. This adjustment ensures that false negative samples are avoided. Consequently, we modify the contrastive loss function to reflect this change and proceed with the training of the models. The results, as shown in Table 11, demonstrate that the models trained with the revised contrastive loss exhibit performance comparable to that of those trained with the original loss. We also plot the contrastive loss curves for several GDPs during the original training process in Fig. 4, all of which exhibit smooth trajectories. These results suggest that the influence of false negative samples on model performance is minimal.

Table 11: Experimental results on the modified and original contrastive loss function. The table presents the GDP-solving accuracy (%) with confidence intervals ($\alpha = 0.05$) for the graph models. 'Graph/SAT Model+ConRep4CO+Modified Loss' denotes training with the modified contrastive loss. 'SAT Back.' refers to SAT model backbone, and 'Graph Back.' denotes graph model backbone.

| SAT Back. | Graph Back. | Difficulty | Model | k-Clique | k-Domset | k-Vercov | k-Color | k-Indset | Matching | Automorph | Overall |
|---|---|---|---|---|---|---|---|---|---|---|---|
| LCG+NeuroSAT | GCN | Easy | Graph Model+ConRep4CO+Modified Loss | 77.1±0.3 | 57.9±0.2 | 61.5±0.3 | 88.7±0.1 | 64.2±0.4 | 71.5±0.2 | 64.4±0.4 | 69.3 |
| | | | Graph Model+ConRep4CO | **79.3±0.3** | **62.0±0.1** | **67.3±0.2** | **90.2±0.1** | **67.5±0.5** | **71.7±0.3** | **65.4±0.3** | **71.9** |
| | | Medium | Graph Model+ConRep4CO+Modified Loss | 70.7±0.3 | 63.0±0.4 | 61.2±0.3 | 79.8±0.4 | 58.9±0.2 | 72.4±0.2 | 63.7±0.4 | 67.1 |
| | | | Graph Model+ConRep4CO | **71.3±0.5** | **64.6±0.2** | **63.3±0.3** | **82.2±0.2** | **64.0±0.1** | **72.8±0.4** | **65.7±0.4** | **69.1** |

Table 12: Experimental results across two graph models under different training methods. 'Graph Model (fully-trained)' refers to the graph model that was trained from scratch with full training data. 'Graph Model+ConRep4CO (fine-tuned)' refers to the fine-tuned graph model after pre-training by ConRep4CO on small datasets.

| Model | k-Clique | k-Domset | k-Vercov | k-Color | k-Indset | Matching | Automorph | Overall |
|---|---|---|---|---|---|---|---|---|
| Graph Model (fully-trained) | 67.3±0.4 | 66.7±0.2 | 65.4±0.4 | 79.1±0.2 | 59.1±0.3 | 72.4±0.1 | 65.4±0.2 | 67.9 |
| Graph Model+ConRep4CO (fine-tuned) | **67.9±0.2** | **67.0±0.1** | **66.6±0.4** | **79.4±0.2** | **61.5±0.4** | **72.6±0.1** | **65.7±0.2** | **68.7** |

Table 13: GDP solving accuracy (%) with confidence intervals ($\alpha = 0.05$) of the graph models on Easy datasets. The 'Overall' column represents the average accuracy across all datasets.

| Model | k-Clique | k-Domset | k-Vercov | k-Color | k-Indset | Matching | Automorph | Overall |
|---|---|---|---|---|---|---|---|---|
| Graph Model-FullData | 77.8±0.1 | 59.0±0.2 | 61.4±0.3 | 87.8±0.4 | 63.2±0.1 | 71.3±0.1 | 64.3±0.3 | 69.3 |
| Graph Model+ConRep4CO | **79.3±0.3** | **62.0±0.1** | **67.3±0.0** | **90.2±0.1** | **67.5±0.1** | **71.7±0.0** | **65.4±0.4** | **71.9** |

### F.2 Further Study on Graph Model Generalization to Large-Scale Data

To further assess the generalization ability of our graph models, we generate large-scale instances for each GDP, with instance sizes ranging from 7 to 20 times larger than those used during pre-training. We then fine-tune the pre-trained models on this large-scale data, using a subset comprising $\frac{1}{8}$ of the training data. We compare the performance of the fine-tuned models with those trained from scratch with full training data, and the results are presented in Table 12, indicating that models pre-trained on smaller instances using ConRep4CO can generalize effectively to larger instances through fine-tuning.

### F.3 Further Study on Data Volume

During the pre-training phase, ConRep4CO utilizes all instances from different domains, while the baselines only have access to the instances from their single domain. To further assess the impact of data volume, we increase the number of training instances for baselines, and train 7 baseline graph models with the GCN backbone separately, each on $7 \times 160{,}000$ graph instances generated from a single problem type, denoted as **Graph Model-FullData**, and compare their performance with our approach in Table 13. It proves that the improved performance is not from increased training data.

## G Large Language Model Usage

In this paper, large language models (LLMs) are only used to find and correct grammatical errors.

