# OpenReview forum: "ConRep4CO: Contrastive Representation Learning of Combinatorial Optimization Instances across Types"
_ICLR.cc/2026/Conference — ICLR 2026 Poster_

### Official Review · Reviewer_nnWe · 2025-10-20

**Soundness:** 3
**Presentation:** 2
**Contribution:** 3
**Rating:** 4
**Confidence:** 2

**Summary:**

This paper considers a contrastive learning based approach to graph Combinatorial Optimization (CO) problems. The approach is based
on associating each CO problem with a graph decision problem. It then associates each GDP with a Boolean Satisfiability problem (SAT) and considers the corresponding SAT graph. The authors then develop models for each of the GDPs and one unified model for the SAT graphs. During training, they use a contrastive loss that purportedly aligns the GDP and SAT modalities they also use a warm-start training technique.

Experimentally, the authors consider seven GDPs include k-clique, k-coloring and k-independent sets. They that adding in the ConRep4CO framework (which appears to refer to their contrastive loss between the SAT and the GDP) improves performaces verses an ablated model. The performance is quantified both on the performance of the representation for solving the GDP and the underlying CO problem, although the performance increase is substantially more modest in the former case (which makes it hard to gain intuition about the larger performance boost in the CO problem).

**Strengths:**

Method does seem to improve performance in most cases

**Weaknesses:**

The method is not clearly explained. More specifically, more background should be given on how a GDP may be related and how the GDP is related to a SAT or a SAT graph. Furthermore, it is not clear how the model actually operates. The overall architecture is not well-explained in the text and in my summary I am making educated guesses based on Figure 2, which is not a substitute for actually explaining the model.

Methodological contributions seem incremental, although this is hard to assess to the general lack of clear explanations as noted above

Additionally, there insufficient discussion of computational complexity / run time. This makes it difficult to weigh the increase in model performance against increased cost. Notably, in CO problems, this is particularly essential. All of these problems are exactly solvable, e.g, by using GUROBI solvers, so the entire name of the game is developing fast, approximate solutions.

The code is not available. This is particularly problematic because a) the method is motivated primarily by experimental results and b) there is not enough detail given on the model in order to be able to reproduce these experiments.

Minor:

The first two sentences of the abstract seem to contradict one another

"texts" should be "text" in line 42

Notational inconsistency G_i^j vs G_j in the start of section 3.3.1

The relative contributions of the SAT backbone and GNN backbone are hard to parse in Table 1

**Questions:**

what is the computational complexity of the proposed model?

how does performance compare to non message-passing based approached, e.g., reinforcement learning

---

> ### Author Response · Authors · 2025-11-18
> **Response to Reviewer nnWe (Part 1/2)**
>
> We are deeply grateful for the time and effort you have invested in reviewing our submission. Your insightful comments are greatly appreciated. Below we provide a point-by-point response to the questions and concerns raised.
>
>
> > **More explanations of our method.**
>
> ### **1. Relationship between GDPs, CO Problems, and SAT Graphs**
>
> A GDP is derived from its corresponding CO problem by introducing a decision threshold $k$. For example, the Maximum Independent Set (MIS) problem is transformed into the $k$-Independent Set problem, which asks whether an independent set of size $k$ exists. This preserves the core structure of the original CO problem.
>
> GDPs are then deterministically transformed into SAT instances (in Conjunctive Normal Form), using established tools like CNFGen [1]. These SAT instances are subsequently represented as graphs (e.g., using Literal-Clause or Variable-Clause Graphs), with bipartite structures where variable/literal vertices are connected to the clause vertices they appear in. This entire pipeline from GDP to SAT graph is rule-based and deterministic.
>
> ### **2. Training Paradigm and Contrastive Learning Mechanism**
>
> As correctly noted, our framework involves training individual models for each GDP alongside a unified model for SAT graphs. The training starts with a warm-start phase, where each model learns foundational task-specific representations independently, followed by a contrastive alignment phase where a contrastive loss is applied simultaneously between all GDP models and the SAT model. This principled mechanism enables effective information transfer and representation enhancement across domains. Below we provide a detailed explanation of this process.
>
> The contrastive objective uses pairs constructed within SAT instances from each single problem domain. The high ratio of negative samples forces the model to distinguish between instances from the same domain, filtering out superficial domain-specific features. As this process is repeated across multiple domains during training, the models naturally converge towards learning universally valuable combinatorial features encoded in the canonical SAT space. This creates a shared representational foundation that facilitates effective information transfer, even though no explicit cross-domain pairs are used.
>
> ### **3. Methodological Contribution**
>
> Our primary contribution is a novel training methodology for representation learning, which is not incremental. While the majority of existing research focuses on the development of specialized solving algorithms, our work shifts the focus toward enhancing model representations through a novel multi-task learning paradigm. Specifically, our approach improves the capability of models to capture transferable problem features, leading to better performance using the same model architecture and solving algorithm.
>
> In contrast to previous multi-task learning approaches such as [2] and [3], which primarily concentrate on architectural design and simply mix data from all tasks during training, we introduce a training methodology that actively leverages multi-task data in a principled manner to achieve mutual enhancement across tasks. Our method does not merely combine datasets; it explicitly facilitates cross-task knowledge transfer through a carefully designed alignment mechanism. We have included comparative experiments with [2] and [3] in **Appendix G.7**, which further prove the effectiveness of our method.
>
> ### **4. Modest Performance Increase for GDP Solving**
>
> We thank the reviewer for insightfully raising this point. We hypothesize that the modest performance gains on GDP solving are partly due to using generic GNN backbones with limited expressive power for specific problems. To test this, we supplement an experiment by replacing the generic GNN backbone with OptGNN [4] for the $k$-Vertex Cover problem, which is specialized for this problem. When enhanced with our training method (by adding alignment with our pre-trained SAT model), the solving accuracy (Medium) significantly improves from **86.4%** to **99.2%**.
>
>
> [1] Cnfgen: A generator of crafted benchmarks.
>
> [2] UniCO: On Unified Combinatorial Optimization via Problem Reduction to Matrix-Encoded General TSP.
>
> [3] GOAL: A Generalist Combinatorial Optimization Agent Learner.
>
> [4] Are graph neural networks optimal approximation algorithms?

---

> ### Author Response · Authors · 2025-11-18
> **Response to Reviewer nnWe (Part 2/2)**
>
> > **Elaboration on computational complexity.**
>
> **Training cost**: We wish to highlight that a discussion of training computational cost is provided in **Appendix F.2** in the original submission. Empirically, on the same hardware, the pre-training of the SAT and graph models via our method requires a computational budget comparable to training the baseline SAT and graph models separately. While this pre-training phase is more substantial than the subsequent fine-tuning for a downstream task, we would like to emphasize that the pre-training phase involving training multiple models is a one-time, upfront cost. When applying to new downstream tasks—including unseen problem domains—only an efficient alignment step with the fixed, pre-trained SAT model is required, not a new full training cycle.
>
> **Inference cost**: During inference, our method introduces no additional computational overhead, as it does not alter the network architecture or the solving algorithm of the specific downstream solvers.
>
>
> > **Available source code.**
>
> We would like to note that we have provided the code for pre-training the SAT and graph models in the original supplementary materials. One can use this code to reproduce the pre-training of SAT and graph models on multiple GDPs, as well as the pre-training on specific downstream tasks. To train and evaluate on downstream tasks, one can directly load the checkpoints after pre-training into the corresponding task-specific repositories (for example, into MVC available at https://github.com/penlu/bespoke-gnn4do [4]), keeping all training and evaluation settings consistent with the original implementations.
>
> For the reviewer's convenience, we have included our pre-trained checkpoint for the downstream MVC task, along with the associated training and evaluation code, at the following anonymous link:
> https://anonymous.4open.science/r/ConRep4CO-OptGNN-034F
>
>
>
> [4] Are graph neural networks optimal approximation algorithms?
>
>
> > **Explanation of the first two sentences in the abstract.**
>
> The two sentences can be interpreted as: while considerable progress has been made in ML4CO, the research focus has predominantly been on the development of specialized solving algorithms or networks. In contrast, relatively little attention has been paid to representation learning for CO problems, particularly to learning representations across different problem domains.
>
>
> > **Typos.**
>
> We thank the reviewer for finding these typos. We have corrected the typos and have also revised the notation, changing $G_{j}$ to $G_{i}^{j}$ for clarity and consistency.
>
>
> > **Elaboration on the SAT backbone and GNN backbone in Table 1.**
>
> We would like to clarify that our proposed method is architecture-agnostic. The purpose of employing various SAT and graph backbones in Table 1 is to demonstrate that our training methodology is effective across a wide range of backbone architectures.
>
> Should a more detailed analysis of backbone influence be helpful, we observe that our method provides consistent improvements across various GNN backbones. Notably, GraphSAGE appears to be particularly well-suited to our framework, yielding especially significant gains. Regarding the SAT backbone, when the GNN backbone is held constant, using NeuroSAT generally leads to greater improvement than using GCN. This is an intuitive result, as NeuroSAT is specifically architected for SAT reasoning, while GCN serves as a more general-purpose operator. Consequently, the performance gain is positively correlated with the expressivity of the SAT backbone, which aligns well with expectations.
>
>
> > **Comparison to non message-passing based approach, e.g., rl.**
>
> We would like to clarify that our work proposes a representation learning method aimed at obtaining a well-initialized model with enhanced representations, rather than a standalone solving algorithm. Therefore, a direct comparison with dedicated solving algorithms may not fully reflect our contribution.
>
> Should our interpretation of your request be incorrect, we would greatly appreciate further clarification on the specific aspects you would like us to address. We would be pleased to provide additional explanations or conduct feasible experiments to better respond to your concerns.
>
>
> Furthermore, to facilitate a more comprehensive understanding of our approach, we have included extensive supplementary experiments in **Appendix G.6** through **G.10** of the submission for your reference.
>
> -----
>
> ## **Thank you again for your attention!**
>
> We sincerely hope that our clarifications and the additional experimental results adequately address your concerns. Your valuable feedback has been instrumental in helping us improve the clarity and completeness of our work. Should you have any further questions or require additional clarification, we would be more than happy to provide it. Thank you once again for your thoughtful review.

---

> ### Author Response · Authors · 2025-11-25
> **Looking forward to your feedback!**
>
> Dear Reviewer nnWe,
>
> Thank you again for your thoughtful review and for the time you've dedicated to our work. We have added clarifications and experiments to resolve your concerns. The code is also available. As the discussion period comes to a close, we wanted to follow up and see if our responses have fully addressed your concerns. We are looking forward to hearing from you.
>
> Thank you again for your attention.
>
> Best regards,
>
> Authors

---

> ### Author Response · Authors · 2025-11-27
> **Follow-up on Our Rebuttal**
>
> Dear Reviewer nnWe,
>
> Thank you once again for your thoughtful review and for raising points that have helped us improve our work. We have submitted our point-by-point responses as well as available code according to your suggestion.
>
> We are pleased to note that the other reviewers have expressed satisfaction with our responses. As the discussion period concludes, we wanted to kindly follow up to see if our responses have adequately addressed your concerns.
>
> We remain fully committed to addressing any further questions you may have. We are readily available to provide clarifications or, if feasible, perform additional experiments to your satisfaction. We would be grateful if you could reconsider your assessment. Thank you again for your time and consideration.
>
> Best regards,
>
> Authors

---

### Official Review · Reviewer_QCGp · 2025-10-27

**Soundness:** 2
**Presentation:** 3
**Contribution:** 2
**Rating:** 4
**Confidence:** 3

**Summary:**

This paper proposes ConRep4CO, a contrastive pre-training framework for learning representations across different combinatorial optimization (CO) problems. The key idea is to transform graph decision problems (GDPs) into SAT form and use contrastive learning to align representations. Positive pairs consist of a GDP instance and its SAT transformation; negative pairs are non-corresponding instances. The method shows improvements on 7 GDPs and 4 downstream optimization problems (MVC, MIS, MC, MDS).

**Strengths:**

- The use of SAT as a common modality to bridge different CO problems is creative and theoretically grounded. The motivation that NPC problems can be reduced to one another through SAT is compelling and elegantly exploited.
- Extensive experiments across 7 GDPs with multiple architectures (GCN, GraphSAGE, PGN, GraphGPS), different difficulty levels; demonstrates generalization to both larger problems and unseen problem types.
- The downstream results on MVC, MIS, MC, and MDS show substantial improvements
- Ablations on cross-domain information transfer, different loss functions, and sensitivity analysis provide good insight into what drives the performance gains.

**Weaknesses:**

- Experiments use very small graphs. Performance gains decrease substantially with scale. Unclear how this scales.
- No empirical comparison to GOAL, UniCO methods despite discussing them. Downstream comparisons only include problem-specific baselines.
- Warm-start strategy mentioned as important but not ablated
- False negatives issue: Modified loss without false negatives shows comparable performance, questioning what contrastive loss actually learns
- Temperature differs between difficulty levels without justification or sensitivity analysis

**Questions:**

- Pre-training uses very small graphs (5-25 vertices) while downstream performance degrades substantially with scale. Real-world CO problems typically involve thousands to tens of thousands of nodes. To be practically useful, would the approach require pre-training on much larger problem instances? Can you provide a guess of the time and ressources needed to train on graphs with 10,000+ vertices, and maintain similar performance improvements at that scale?
- Can you add empirical comparisons to GOAL and UniCO ? These seem like important missing baselines.
- How sensitive is performance to warm-start? What happens without it?
- SAT Model Failure: Why does SAT model show 50% accuracy (random) on many hard instances while graph models transfer reasonably?
- Temperature Selection: Why does temperature differ between easy/medium datasets? How was this chosen?

---

> ### Author Response · Authors · 2025-11-18
> **Response to Reviewer QCGp (Part 1/3)**
>
> We are deeply grateful for the time and effort you have invested in reviewing our submission. Your insightful comments are greatly appreciated. Below we provide a point-by-point response to the questions and concerns raised.
>
>
> > **Experiments on large graphs.**
>
> To further investigate the phenomenon of diminishing performance gains with increasing problem scale, as observed when models are pre-trained on easy-level instances from a specific downstream task, we conduct a series of experiments on large-scale problems using pre-training data of varying difficulty levels.
>
> We evaluate our approach on MVC tasks using ER graphs of increasing scale: ER(600, 1000), ER(1000, 2000), and ER(2000, 3000), as well as on a very large-scale MIS task using ER(9000, 11000) graphs.
>
> For each task, we pre-train models using easy-, medium-, and hard-level instances from the respective problem domain, corresponding to the configurations **Ours-Easy**, **Ours-Medium**, and **Ours-Hard**. Note that **Ours-Easy** represents the method used in the main text. The results are presented in the table below, where values in parentheses indicate the performance gain over the baseline, calculated consistently with the method outlined in the paper.
>
>
> |Problem|Graph|Optimal|Baseline|Ours-Easy|Ours-Medium|Ours-Hard|
> |-|-|-|-|-|-|-|
> |MVC|ER(600,1000)|798.54|806.81|805.36 (17.53%)|804.83 (23.94%)|803.99 (34.10%)|
> |MVC|ER(1000,2000)|1320.78|1331.19|1329.75 (13.83%)|1328.48 (26.03%)|1327.90 (31.60%)|
> |MVC|ER(2000,3000)|2476.30|2498.79|2496.46 (10.36%)|2493.87 (21.88%)|2492.07 (29.88%)|
> |MIS|ER(9000,11000)|381.31|356.47|358.73 (9.10%)|360.36 (15.66%)|362.11 (22.71%)|
>
>
> The results show that even the **Ours-Easy** configuration maintains a non-trivial gain of approximately 10% on large-scale problems, demonstrating its residual generalization capability.
>
> Furthermore, increasing the scale of the pre-training data significantly mitigates the diminishing gain trend. Notably, effective mitigation does not require pre-training on graphs of massive scale (e.g., with 10,000 vertices). Pre-training on our hard-level instances (comprising graphs with only up to 25 vertices) is sufficient to achieve stable and substantial gains, maintaining an improvement of around 20% even on the very large ER(9000, 11000) instances.
>
> These results demonstrate the potential of our approach for enhancing real-world, large-scale problem-solving. Additional details are available in **Appendix G.8**.

---

> ### Author Response · Authors · 2025-11-18
> **Response to Reviewer QCGp (Part 2/3)**
>
> > **Empirical comparison to GOAL, UniCO.**
>
> Our method introduces an innovation in the training approach, which is architecture-agnostic. Therefore, when comparing with GOAL and UniCO, our objective is to evaluate the relative efficacy of our proposed training strategy against their multi-task training methodology. To ensure a fair comparison, we use the same network architectures as the original works—namely, MatPOENet for UniCO and the GOAL framework for its respective tasks.
>
> We select representative tasks from each work: ATSP and 2DTSP from UniCO, and ATSP, MVC, and MIS from GOAL. For each, we compare three training strategies:
>
> 1. **-Single**: The model is trained exclusively on the single target task.
> 2. **-Mixed-Tuned** or **-Multi-Tuned**: The model is first trained on a mixture of multiple tasks and then fine-tuned on the single target task. The fine-tuning dataset matches the size of the single-task training set to isolate the effect of the training strategy from data volume.
> 3. **Ours**: The model is first aligned with our pre-trained SAT model using a small amount of task-specific instances, and then trained following the same procedure and data amount as the single-task setting.
>
> Consistent with the original papers, we report the average tour length for MatPOENet and the average gap for GOAL in the tables below.
>
>
> |Model|ATSP20|ATSP50|ATSP100|2DTSP20|2DTSP50|2DTSP100|
> |-|-|-|-|-|-|-|
> |MatPOENet-Single|1.5784|1.5864|1.6139|3.8427|5.7345|8.0972|
> |MatPOENet-Mixed-Tuned|1.5778|1.5870|1.6143|3.8419|5.7342|8.1007|
> |Ours|**1.5692**|**1.5809**|**1.6098**|**3.8368**|**5.7296**|**8.0931**|
>
> |Model|ATSP100|MVC100|MIS100|
> |-|-|-|-|
> |GOAL-Single|0.32%|0.21%|0.16%|
> |GOAL-Multi-Tuned|0.30%|0.22%|0.15%|
> |Ours|**0.25%**|**0.17%**|**0.13%**|
>
> The results indicate that simply pre-training on a mixture of data from multiple problems followed by extensive single-task fine-tuning does not yield significant improvements over direct single-task training.
>
> The primary value of these baseline methods (GOAL and UniCO) lies in their ability to accommodate different problems within a single model, achieving performance close to that of specialized models after light fine-tuning. However, they do not enhance single-task performance beyond what is achieved by dedicated single-task training.
>
> In contrast, our method is designed to exploit commonalities across different problems through contrastive learning, thereby improving the learned representations and ultimately boosting performance on individual tasks. Additional details are available in **Appendix G.7.1** and **Appendix G.7.2**.
>
>
> > **Ablation study with warm-start strategy.**
>
> We have conducted an ablation study on the warm-start phase by training a model that skips this phase, denoted as **w/o warm-start**. The results are presented in the table below.
>
> |Model|k-Clique|k-Domset|k-Vercov|k-Color|k-Indset|Matching|Automorph|Overall|
> |-|-|-|-|-|-|-|-|-|
> |w/o warm-start|71.4|61.9|65.3|90.4|71.1|65.9|62.6|69.8|
> |with warm-start|**79.7**|**63.2**|**70.8**|**93.3**|**75.3**|**71.0**|**63.9**|**73.9**|
>
> As shown, the omission of the warm-start phase leads to a noticeable performance decline. We attribute this to the fact that applying contrastive learning at the very beginning can disrupt the model's acquisition of task-specific representations. Consequently, the information transferred in subsequent phases may incorporate greater noise or bias, which adversely impacts representation quality. Additional details are available in **Appendix G.9**.
>
>
> > **Modified loss without false negatives shows comparable performance.**
>
> The presence of false negative samples is not a predetermined or known condition. In fact, within the vast solution space of SAT problems, even minor perturbations can lead to significant alterations in the problem's structure. Furthermore, our data generation process explicitly ensures distinct graph structures from the source domains. Under these conditions, the likelihood of encountering false negative samples is highly improbable. Our experimental results in Table 12 aim to demonstrate that, even under the assumption that such samples exist, their impact on the overall model performance is minimal.

---

> ### Author Response · Authors · 2025-11-18
> **Response to Reviewer QCGp (Part 3/3)**
>
> > **Further analysis on temperature selection.**
>
> In our experiments, the temperature hyperparameter is initially selected based on convergence speed. To provide a more comprehensive understanding of its impact, we have conducted an ablation study across a range of temperature values. The results are presented in the table below, where "Converged Epoch" indicates the training epoch at which the contrastive loss plateaus.
>
> Easy:
> |Temperature|k-Clique|k-Domset|k-Vercov|k-Color|k-Indset|Matching|Automorph|Overall|Converged Epoch|
> |-|-|-|-|-|-|-|-|-|-|
> |0.1|79.7|63.2|70.8|93.3|75.3|71.0|63.9|73.9|16|
> |0.2|79.5|63.0|70.8|93.2|75.0|70.7|63.9|73.7|18|
> |0.3|79.6|63.2|70.7|93.2|75.2|71.1|63.7|73.8|20|
> |0.4|79.7|63.1|70.8|93.5|75.5|71.2|63.9|74.0|19|
> |0.5|79.7|63.1|70.6|93.1|75.4|70.9|63.9|73.8|19|
> |0.6|79.9|63.1|70.9|93.1|75.4|71.0|64.2|73.9|20|
> |0.7|79.7|63.2|70.9|93.3|75.4|71.0|63.8|73.9|22|
> |0.8|79.6|63.2|70.8|93.2|75.1|71.0|63.6|73.8|24|
>
> Medium:
> |Temperature|k-Clique|k-Domset|k-Vercov|k-Color|k-Indset|Matching|Automorph|Overall|Converged Epoch|
> |-|-|-|-|-|-|-|-|-|-|
> |0.1|72.6|64.2|66.5|86.0|70.0|71.7|65.0|70.9|24|
> |0.2|72.4|64.5|66.6|85.8|70.2|71.5|64.8|70.8|23|
> |0.3|72.9|64.0|66.8|85.9|69.9|71.6|64.7|70.8|21|
> |0.4|72.7|64.0|66.7|86.0|70.4|72.0|64.9|71.0|18|
> |0.5|72.8|64.1|66.7|85.9|70.1|71.7|64.8|70.9|18|
> |0.6|72.7|64.0|66.7|85.7|70.0|71.8|64.7|70.8|20|
> |0.7|72.5|64.1|66.5|85.8|70.0|71.6|64.6|70.7|23|
> |0.8|72.6|64.1|66.5|85.9|70.1|71.6|64.8|70.8|24|
>
> The results demonstrate that while the choice of temperature influences the convergence rate, the final performance metrics remain highly consistent. Additional details are available in **Appendix G.10**.
>
>
> > **SAT model fails while graph model does not.**
>
> The performance difference stems from the distinct fine-tuning procedures used for the SAT model versus the graph models. The graph models are fine-tuned on individual problem instances, whereas the SAT model is fine-tuned concurrently across all seven problem domains.
>
> It is important to note that, since our test dataset is balanced with 50% positive and 50% negative labels, an exact 50% accuracy does not indicate random guessing but rather a model that consistently predicts only one class (either all 0s or all 1s). We observe that this behavior primarily occurs when the SAT model employs a GCN backbone, which appears to introduce a representational bias on hard instances.
>
> Although the GCN SAT model itself exhibits limited generalization on hard instances of certain problem types, it remains effective on in-distribution tasks and facilitates beneficial information transfer across different graph models. This transfer enhances learned representations but may also introduce some bias. In some domains, this enhanced representation allows the graph model to mitigate the bias during its fine-tuning stage, leading to successful performance. In other, more challenging domains like $k$-DomSet, this bias proves more difficult to overcome.
>
> To demonstrate that single-domain fine-tuning can help alleviate this bias, we separately fine-tune the VCG+GCN SAT backbone on 20,000 easy-level instances from each individual domain and evaluate its generalization performance on hard instances. The results are presented in the table below.
>
> Hard:
>
> |Model|k-Clique|k-Domset|k-Vercov|k-Color|k-Indset|Matching|Automorph|Overall|
> |-|-|-|-|-|-|-|-|-|
> |SAT Model|63.2|77.5|69.6|61.7|65.4|79.1|79.2|70.8|
> |SAT Model+ConRep4CO|68.4|79.0|74.2|68.9|71.1|80.2|79.4|74.5|
>
> Whereas the backbone fails to generalize when fine-tuned jointly across all seven domains, fine-tuning on each domain individually successfully prevents this generalization failure. This result confirms that single-domain fine-tuning is an effective strategy for mitigating the bias introduced during cross-domain pre-training.
>
> -----
>
> ## **Thank you again for your attention!**
>
> We sincerely hope that our clarifications and the additional experimental results adequately address your concerns. Your valuable feedback has been instrumental in helping us improve the clarity and completeness of our work. Should you have any further questions or require additional clarification, we would be more than happy to provide it. Thank you once again for your thoughtful review.

---

> > ### Comment · Reviewer_QCGp · 2025-11-20
> >
> > Thanks a lot for the clarifications and apologies for the delay. I appreciate you taking the time to answer precisely to my questions. In light of these results, I raise my score.

---

> > > ### Author Response · Authors · 2025-11-21
> > > **Thank you for your timely reply!**
> > >
> > > We are profoundly grateful for the time and care you have taken in reviewing our submission and response. Your support and positive assessment are immensely encouraging. Your insightful feedback has been instrumental in refining our work, and we are truly honored by your recommendation.

---

### Official Review · Reviewer_FRhi · 2025-11-01

**Soundness:** 2
**Presentation:** 2
**Contribution:** 3
**Rating:** 6
**Confidence:** 4

**Summary:**

This paper introduces ConRep4CO, a framework that leverages contrastive pretraining to generate representations for combinatorial optimization. The core innovation lies in its method of unifying diverse NP-complete problems by converting them into a Boolean satisfiability format; these SAT-derived versions serve as positive counterparts to their original instances within the contrastive learning objective. Empirical evaluation across seven graph-based decision problems and four optimization tasks indicates that the framework not only enhances performance but also exhibits robust generalization, positioning it as a viable paradigm for unified representation learning in the CO domain.

**Strengths:**

1. The core novelty of this work lies in its use of Boolean satisfiability as a unifying representational bridge, enabling knowledge transfer across distinct combinatorial optimization problems.

2. The experiments are sufficient, rigorously evaluating the framework across various graph decision problem categories, architectural backbones, and downstream solver methodologies.

3. The results demonstrate observable empirical improvements, confirming that the method enhances both the quality of the learned representations and the end-to-end performance of the solvers that utilize them.

**Weaknesses:**

1. While the results are compelling, the paper would benefit from a deeper mechanistic analysis of the SAT-based alignment process. A more thorough investigation into why this specific form of contrastive learning fosters such effective cross-domain generalization would strengthen the theoretical contribution.

2. It seems that the framework is explicitly designed for problems that can be efficiently reduced to SAT (primarily NP-complete problems). I am not sure whether I missed anything. But its applicability to problems outside this class, or to CO problems with complex, non-logical constraints (e.g., certain scheduling or routing constraints), is unclear and not explored.

3. The practical scalability of ConRep4CO warrants further discussion, as the computational complexity associated with the CNF transformation and the requirement to train multiple models could present significant bottlenecks for larger, real-world instances.

**Questions:**

1. The paper justifies the use of SAT through its universal expressiveness (reducibility). However, a deeper theoretical or visualized explanation of its representational advantage would be beneficial. For instance, does the SAT form act as an information bottleneck, stripping away problem-specific noise to reveal a common core, or does it function as a canonical encoder that explicitly captures fundamental combinatorial structures shared across different GDPs?

2.  To ensure reproducibility and provide deeper insight, could the authors elaborate on the training details? Specifically, clarifying the objective and impact of the warm-start phase, and explaining how the balance (relative weighting) between the two loss components was determined or tuned would be highly valuable.

3. The premise of the framework is that diversity in pre-training tasks fosters generalization. Was this correlation investigated? Specifically, does performance on downstream tasks improve when the model is pre-trained on a broader variety of problem types? Furthermore, which specific design elements (e.g., the specific contrastive pairing, the SAT transformation itself) are identified as most critical for enabling this effective cross-domain knowledge transfer?

4. How does the method scale with an increasing number of pre-training problem types? Have you considered or experimented with other unifying formalisms beyond SAT (e.g., Integer Linear Programming) that might offer different trade-offs in terms of representation compactness or transformation complexity?

---

> ### Author Response · Authors · 2025-11-18
> **Response to Reviewer FRhi (Part 1/3)**
>
> We are deeply grateful for the time and effort you have invested in reviewing our submission. Your insightful comments are greatly appreciated. Below we provide a point-by-point response to the questions and concerns raised.
>
>
> > **A deeper mechanistic analysis of the SAT-based alignment process would be beneficial.**
>
> We thank the reviewer for this insightful comment. Our interpretation is that the SAT transformation's power lies in its ability to **map diverse problems into a unified constraint-based space where their fundamental structural similarities become explicit and accessible to the model**.
>
> This process is not merely a syntactic change but a re-representation that highlights shared combinatorial challenges (e.g., conflicts, dependencies, satisfiability) common to different problems. The subsequent contrastive learning operates within this unified space.
>
> While our contrastive objective uses positive and negative pairs constructed within SAT instances from a single problem domain, the sheer prevalence of negative pairs (which far outnumber the positives) plays a crucial role. By aggressively pushing apart the representations of instances from the same domain, the model is forced to filter out superficial, domain-specific features.
>
> Through multi-domain training, this process naturally converges toward identifying and strengthening universally valuable features—the essential combinatorial building blocks shared across domains. The key cross-domain benefit emerges because models from different domains are all optimized within the same canonical SAT space. Through multi-domain training, they arrive at similar robust representations of fundamental combinatorial structures, creating a shared representational foundation that facilitates effective information transfer. This explains why our method enhances performance on individual tasks despite using only intra-domain supervision during contrastive learning.
>
>
> > **Application to broader CO problems. Have you considered or experimented with other unifying formalisms beyond SAT (e.g., Integer Linear Programming)?**
>
> Your understanding is correct that our method utilizes transformations from specific problems to SAT. We would like to emphasize that the class of problems amenable to this approach is non-trivial and practically significant, encompassing challenging real-world scenarios. To explicitly explore the applicability of our method to a broader range of problems, we conduct experiments on two general TSP tasks—ATSP and 2DTSP—drawn from [1]. We adopted the methodology from [1] to accomplish the transformation between general TSP (HCP) instances and SAT (3-SAT) instances, which subsequently serve as the pre-training data for the specific tasks. The results, reported in terms of average tour length (lower is better), are presented below:
>
> |Model|ATSP20|ATSP50|ATSP100|2DTSP20|2DTSP50|2DTSP100|
> |-|-|-|-|-|-|-|
> |MatPOENet-Single|1.5784|1.5864|1.6139|3.8427|5.7345|8.0972|
> |MatPOENet-Mixed-Tuned|1.5778|1.5870|1.6143|3.8419|5.7342|8.1007|
> |Ours|**1.5692**|**1.5809**|**1.6098**|**3.8368**|**5.7296**|**8.0931**|
>
> As shown, our method achieves competitive performance on these general TSP tasks, outperforming the multi-task learning baseline. Additional details are available in **Appendix G.7.1**.
>
> Regarding the use of other unifying formalisms, such as Integer Linear Programming (ILP), we recognize their generality but also note a significant challenge: their extreme expressiveness results in a vastly larger and more diffuse representation space. In such spaces, the structural relationships between different source domains can become less distinct, potentially requiring a much larger number of problem domains to effectively capture meaningful commonalities. This would, in turn, introduce substantial computational complexity and training burden. We believe that investigating trade-offs between representational compactness and transformation complexity is a challenging yet promising research direction, and we plan to explore this as part of our future work.
>
> [1] UniCO: On Unified Combinatorial Optimization via Problem Reduction to Matrix-Encoded General TSP.

---

> ### Author Response · Authors · 2025-11-18
> **Response to Reviewer FRhi (Part 2/3)**
>
> > **Training detail: ablation study on warm-start.**
>
> We have conducted an ablation study on the warm-start phase by training a model that skips this phase, denoted as **w/o warm-start**. The results are presented in the table below.
>
> |Model|k-Clique|k-Domset|k-Vercov|k-Color|k-Indset|Matching|Automorph|Overall|
> |-|-|-|-|-|-|-|-|-|
> |w/o warm-start|71.4|61.9|65.3|90.4|71.1|65.9|62.6|69.8|
> |with warm-start|**79.7**|**63.2**|**70.8**|**93.3**|**75.3**|**71.0**|**63.9**|**73.9**|
>
> As shown, the omission of the warm-start phase leads to a noticeable performance decline. We attribute this to the fact that applying contrastive learning at the very beginning can disrupt the model's acquisition of task-specific representations. Consequently, the information transferred in subsequent phases may incorporate greater bias, which adversely impacts representation quality. It can be interpreted that the warm start phase ensures the information being shared across different problems to be meaningful. Additional details are available in **Appendix G.9**.
>
>
>
> > **Training detail: balance the two loss components.**
>
> The warm-start phase is guided by the decision loss. Therefore, the coefficient $\beta$ is introduced to prevent this loss from dominating the pre-training objective in the contrastive learning phase, thereby allowing the model to better explore the representation space. Consequently, $\beta$ should be set to a relatively low value. Empirically, we find that values within the range of 0.5 to 1 lead to stable results.
>
>
>
>
> > **The correlation between the diversity in pre-training tasks and the downstream task improvement.**
>
> To further investigate the impact of incorporating multiple domains, we have conducted extensive experiments to analyze how the number of pre-training domains influences downstream performance on MVC, MIS, MC, and MDS tasks. We design seven pre-training configurations, utilizing data from 1 to 7 distinct problem domains for pre-training (denoted as **Ours-1** to **Ours-7**). The results are summarized in the table below, where values in parentheses represent the performance gain over the baseline, calculated consistently with the method outlined in the paper.
>
> |Problem|Graph|Optimal|Baseline|Ours-1|Ours-2|Ours-3|Ours-4|Ours-5|Ours-6|Ours-7|
> |-|-|-|-|-|-|-|-|-|-|-|
> |MVC|ER(50,100)|54.62|55.87|55.75 (9.60%)|55.39 (38.40%)|55.07 (64.00%)|54.76 (88.80%)|54.74 (90.40%)|54.68 (95.20%)|54.70 (93.60%)|
> |MVC|ER(100,200)|122.79|126.04|125.86 (5.54%)|125.10 (28.92%)|124.89 (35.38%)|124.51 (47.08%)|124.44 (49.23%)|124.39 (50.77%)|124.37 (51.40%)|
> |MVC|ER(400,500)|417.42|420.51|420.40 (3.56%)|419.86 (21.04%)|419.63 (28.48%)|419.49 (33.01%)|419.39 (36.25%)|419.33 (38.19%)|419.31 (38.83%)|
> |MIS|RB(200,300)|20.10|19.18|19.22 (4.35%)|19.47 (31.52%)|19.52 (36.96%)|19.55 (40.22%)|19.53 (38.04%)|19.57 (42.39%)|19.56 (41.30%)|
> |MIS|RB(800,1200)|43.15|37.48|37.62 (2.47%)|38.47 (17.46%)|38.66 (20.81%)|38.74 (22.22%)|38.79 (23.10%)|38.77 (22.75%)|38.79 (23.10%)|
> |MC|RB(200,300)|19.05|16.24|16.41 (6.05%)|16.90 (23.49%)|17.23 (35.23%)|17.36 (39.86%)|17.44 (42.70%)|17.47 (43.77%)|17.47 (43.77%)|
> |MC|RB(800,1200)|33.89|31.42|31.49 (2.83%)|31.75 (13.36%)|31.90 (19.43%)|32.07 (26.31%)|32.11 (27.94%)|32.12 (28.34%)|32.14 (29.15%)|
> |MDS|RB(200,300)|27.89|28.61|28.56 (6.94%)|28.42 (26.39%)|28.28 (45.83%)|28.24 (51.39%)|28.18 (59.72%)|28.18 (59.72%)|28.19 (58.33%)|
> |MDS|RB(800,1200)|103.80|110.28|110.04 (3.70%)|109.12 (17.90%)|108.68 (24.69%)|108.34 (29.94%)|108.22 (31.79%)|108.17 (32.56%)|108.19 (32.25%)|
>
> As the number of pre-training domains increases, the gains grow significantly, but in a sublinear fashion. The gains eventually saturate, beyond which further increasing the number of pre-training domains yields diminishing returns. Given the associated growth in computational cost, we find that pre-training with 5-6 domains offers a favorable trade-off. Additional details, including the specific domains used for pre-training in each configuration for each downstream task, can be found in **Appendix G.6**.
>
>
> > **Which specific design elements (e.g., the specific contrastive pairing, the SAT transformation itself) are identified as most critical for enabling this effective cross-domain knowledge transfer?**
>
> As shown in the table above, the SAT transformation itself yields a modest performance gain. The substantial improvement observed with an increasing number of pre-training domains demonstrates the additional benefit brought by contrastive pairing. It is important to note, however, that the contrastive pairing is implemented based on the SAT transformation. The two components are complementary and mutually reinforcing, and it would be inaccurate to attribute greater importance to one over the other.

---

> ### Author Response · Authors · 2025-11-18
> **Response to Reviewer FRhi (Part 3/3)**
>
> > **Discussion on practical scalability: computational complexity of the CNF transformation and the requirement to train multiple models.**
>
> Theoretically, the CNF transformation possesses polynomial-time complexity [2], and therefore does not represent a computational bottleneck in our pipeline.
>
> Regarding the requirement to train multiple models, we would like to clarify that this is only necessary during the initial pre-training phase of the SAT model. When applying to specific downstream tasks—even for unseen problem domains—one only needs to perform alignment with the fixed, pre-trained SAT model, which is efficient. Consequently, the need to train multiple models from scratch is infrequent. Furthermore, our experiments below demonstrate that the pre-training data does not need to be excessively large:
>
> We evaluate our approach on MVC tasks using ER graphs of increasing scale: ER(600, 1000), ER(1000, 2000), and ER(2000, 3000), as well as on a very large-scale MIS task using ER(9000, 11000) graphs.
>
> For each task, we pre-train models using easy-, medium-, and hard-level instances from the respective problem domain, corresponding to the configurations **Ours-Easy**, **Ours-Medium**, and **Ours-Hard**. Note that **Ours-Easy** represents the method used in the main text. The results are presented in the table below, where values in parentheses indicate the performance gain over the baseline, calculated consistently with the method outlined in the paper.
>
> |Problem|Graph|Optimal|Baseline|Ours-Easy|Ours-Medium|Ours-Hard|
> |-|-|-|-|-|-|-|
> |MVC|ER(600,1000)|798.54|806.81|805.36 (17.53%)|804.83 (23.94%)|803.99 (34.10%)|
> |MVC|ER(1000,2000)|1320.78|1331.19|1329.75 (13.83%)|1328.48 (26.03%)|1327.90 (31.60%)|
> |MVC|ER(2000,3000)|2476.30|2498.79|2496.46 (10.36%)|2493.87 (21.88%)|2492.07 (29.88%)|
> |MIS|ER(9000,11000)|381.31|356.47|358.73 (9.10%)|360.36 (15.66%)|362.11 (22.71%)|
>
> Pre-training on our hard-level instances (comprising graphs with only up to 25 vertices) is sufficient to achieve substantial gains, maintaining an improvement of around 20% even on the very large ER(9000, 11000) instances. Additional details are available in **Appendix G.8**.
>
>
> [2] Reducibility Among Combinatorial Problems.
>
>
> -----
>
> ## **Thank you again for your attention!**
>
> We sincerely hope that our clarifications and the additional experimental results adequately address your concerns. Your valuable feedback has been instrumental in helping us improve the clarity and completeness of our work. Should you have any further questions or require additional clarification, we would be more than happy to provide it. Thank you once again for your thoughtful review.

---

> > ### Comment · Reviewer_FRhi · 2025-11-26
> >
> > Thank you for the rebuttal, which addressed some of my concerns. I have decided to remain my score.

---

> > > ### Author Response · Authors · 2025-11-26
> > > **Thank you for your continued positive assessment!**
> > >
> > > We are deeply grateful for your thoughtful review and for the encouraging support you've given our paper. Your constructive feedback was invaluable in its refinement, and we are truly honored by your recommendation.

---

### Official Review · Reviewer_ZxEz · 2025-11-03

**Soundness:** 2
**Presentation:** 2
**Contribution:** 2
**Rating:** 4
**Confidence:** 3

**Summary:**

This paper introduces ConRep4CO, a novel training framework that leverages contrastive learning to significantly enhance the learned representations of graph decision problems. Experimental results demonstrate that ConRep4CO's contrastive learning techniques indeed improve model performance. Furthermore, the approach shows promising extensibility, maintaining performance improvement when applied to unseen combinatorial optimization problems.

**Strengths:**

-	The illustration of the training process is clear and easy to follow.
-	The experimental evaluations are comprehensive.

**Weaknesses:**

-	The definitions of some symbols are missing. What are confidence intervals? What is ER in Table 4? What is RB used in Table 5?
-	It is unclear why the models are only trained on the easy and medium problems without considering the hard problems (Tables 1 and 2).
-	Some numeric calculations in Table 4 are incorrect. For instance, GCNN+ConRep4CO gets 55.17 OBJ score, while the optimal score is 54.62. The gap_{abs} score should be 55.17-54.62=0.55, not 0.57, which is listed in the table. In fact, I find that all the gap_{abs} scores of GCNN+ConRep4CO are incorrect.
-	The improvement introduced by training the model across multiple domains (Table 6) seems to be very small. The difference between the single domain setting and the multiple domain setting is within the standard deviation.

**Questions:**

1.	What are the definitions of confidence intervals, ER, and RB?
2.	Why are models only trained on the easy and medium problems (Tables 1 and 2)?

---

> ### Author Response · Authors · 2025-11-18
> **Response to Reviewer ZxEz (Part 1/2)**
>
> We are deeply grateful for the time and effort you have invested in reviewing our submission. Your insightful comments are greatly appreciated. Below we provide a point-by-point response to the questions and concerns raised.
>
>
> > **Definitions of confidence intervals, ER, RB, and BA.**
>
> We would be happy to provide further clarification on these definitions as needed.
>
> A **confidence interval** provides a range of values that, based on the test data, is highly likely to contain the model's "true" performance on the entire population of data. It provides an estimate of the uncertainty associated with the reported results due to the finite nature of the test set. The calculation of confidence intervals is deterministic once the statistical samples are obtained.
>
> In our experiments, we report the GDP solving accuracy (%) with 95% confidence intervals. Take the value **"79.3±0.3"** as an example, the point estimate (**79.3**) is the average GDP solving accuracy, and the margin of error (**±0.3**) quantifies the uncertainty in the point estimate due to the finite size of the test set. It can be interpreted as: "We are 95% confident that the true performance of our model, if it could be evaluated on the entire, infinite population of data for this problem, would fall between **79.0** and **79.6**."
>
>
> **ER (Erdős–Rényi Model)**, **RB (Model RB)**, and **BA (Barabási–Albert Model)** can all serve as random graph generators to generate the graph instances required in our downstream tasks.
>
> **ER** [1] generates graphs by connecting each pair of nodes with a fixed probability $p$, producing homogeneous, unstructured random graphs ideal for baseline comparisons.
>
> **RB** [2] generates constraint satisfaction problem instances that encode NP-hard graph problems, specializing in creating computationally challenging benchmark instances.
>
> **BA** [3] produces scale-free graphs through growth and preferential attachment, creating realistic graphs with hub structures commonly found in real-world systems.
>
>
> [1] Erdős P, Rényi A. On Random Graphs[J]. Publicationes Mathematicae. 1959, 6: 290–297.
>
> [2] Xu K, Li W. Exact phase transitions in random constraint satisfaction problems[J]. Journal of Artificial Intelligence Research, 2000, 12: 93-103.
>
> [3] Barabási A L, Albert R. Emergence of scaling in random networks[J]. science, 1999, 286(5439): 509-512.
>
>
> > **Why not train on the hard problems in Table 1 and 2?**
>
> In our experiments, hard-level GDP instances are used to measure the generalizability of the learned representations. The results in the original Table 2 report performance from models trained on easy and medium instances and tested on hard instances to evaluate this generalization.
>
> To further strengthen our analysis, we have supplemented experiments trained directly on hard-level GDP instances. For each problem domain, we generate a dedicated dataset of 160,000 hard-level instances for training, with 20,000 for validation and 20,000 for testing. Using the LCG+NeuroSAT+GraphGPS backbone, and maintaining consistency with the training procedure described in our paper, we obtain the results presented in the table below.
>
> |Model|k-Clique|k-Domset|k-Vercov|k-Color|k-Indset|Matching|Automorph|Overall|
> |-|-|-|-|-|-|-|-|-|
> |Graph Model (hard)|66.4|60.2|61.6|80.7|57.4|68.1|60.3|65.0|
> |Graph Model+ConRep4CO (hard)|**69.6**|**68.3**|**74.5**|**82.3**|**66.4**|**68.6**|**63.7**|**70.5**|
>
> Furthermore, we recognize that solving hard-level GDP instances alone may not fully reflect performance on challenging CO problems. Therefore, we also evaluate using instances of different difficulty levels (easy, medium, hard) as pre-training data for downstream tasks at a larger scale. The results, shown in the following table, demonstrate the scalability of our approach. In the table, 'ER(600, 1000)' denotes instances that are ER graphs with 600–1000 nodes, with other entries following the same convention. The values in parentheses represent the performance gain over the baseline, calculated consistently with the method outlined in the paper.
>
> |Problem|Graph|Optimal|Baseline|Ours-Easy|Ours-Medium|Ours-Hard|
> |-|-|-|-|-|-|-|
> |MVC|ER(600,1000)|798.54|806.81|805.36 (17.53%)|804.83 (23.94%)|803.99 (34.10%)|
> |MVC|ER(1000,2000)|1320.78|1331.19|1329.75 (13.83%)|1328.48 (26.03%)|1327.90 (31.60%)|
> |MVC|ER(2000,3000)|2476.30|2498.79|2496.46 (10.36%)|2493.87 (21.88%)|2492.07 (29.88%)|
> |MIS|ER(9000,11000)|381.31|356.47|358.73 (9.10%)|360.36 (15.66%)|362.11 (22.71%)|
>
> Additional details for this experiment are available in **Appendix G.8**.

---

> > ### Comment · Reviewer_ZxEz · 2025-11-26
> >
> > Thank you for addressing my concerns and including additional experiments!
> > I am inclined towards accepting this paper and I will update my score.

---

> > > ### Author Response · Authors · 2025-11-26
> > > **Thank you for raising the score!**
> > >
> > > We want to express our sincere gratitude for your thorough evaluation and insightful feedback on our paper. Your positive assessment and supportive words are immensely encouraging. We are truly honored by your recommendation and are grateful for your help us refining our work.

---

> ### Author Response · Authors · 2025-11-18
> **Response to Reviewer ZxEz (Part 2/2)**
>
> > **Numeric calculation errors.**
>
> We are grateful to the reviewer for identifying the calculation errors in Table 4. We have carefully corrected these calculation errors. A portion of the corrected table is presented here for your verification.
>
> |Graph|GCNN ($\text{gap}_{\text{abs}}$)|GCNN+ConRep4CO ($\text{gap}_{\text{abs}}$)|Gain|
> |-|-|-|-|
> |ER(50, 100)|0.72|0.55|23.61%|
> |ER(50, 100)|5.50|3.96|28.00%|
> |ER(50, 100)|26.01|19.35|25.61%|
> |Avg. Gain|-|-|25.74%|
>
>
> > **Improvment introduced by multiple domains.**
>
> To further investigate the impact of incorporating multiple domains, we have conducted extensive experiments to analyze how the number of pre-training domains influences downstream performance on MVC, MIS, MC, and MDS tasks. We design seven pre-training configurations, utilizing data from 1 to 7 distinct problem domains (denoted as **Ours-1** to **Ours-7**). The results are summarized in the table below, where values in parentheses represent the performance gain over the baseline, calculated consistently with the method outlined in the paper.
>
> |Problem|Graph|Optimal|Baseline|Ours-1|Ours-2|Ours-3|Ours-4|Ours-5|Ours-6|Ours-7|
> |-|-|-|-|-|-|-|-|-|-|-|
> |MVC|ER(50,100)|54.62|55.87|55.75 (9.60%)|55.39 (38.40%)|55.07 (64.00%)|54.76 (88.80%)|54.74 (90.40%)|54.68 (95.20%)|54.70 (93.60%)|
> |MVC|ER(100,200)|122.79|126.04|125.86 (5.54%)|125.10 (28.92%)|124.89 (35.38%)|124.51 (47.08%)|124.44 (49.23%)|124.39 (50.77%)|124.37 (51.40%)|
> |MVC|ER(400,500)|417.42|420.51|420.40 (3.56%)|419.86 (21.04%)|419.63 (28.48%)|419.49 (33.01%)|419.39 (36.25%)|419.33 (38.19%)|419.31 (38.83%)|
> |MIS|RB(200,300)|20.10|19.18|19.22 (4.35%)|19.47 (31.52%)|19.52 (36.96%)|19.55 (40.22%)|19.53 (38.04%)|19.57 (42.39%)|19.56 (41.30%)|
> |MIS|RB(800,1200)|43.15|37.48|37.62 (2.47%)|38.47 (17.46%)|38.66 (20.81%)|38.74 (22.22%)|38.79 (23.10%)|38.77 (22.75%)|38.79 (23.10%)|
> |MC|RB(200,300)|19.05|16.24|16.41 (6.05%)|16.90 (23.49%)|17.23 (35.23%)|17.36 (39.86%)|17.44 (42.70%)|17.47 (43.77%)|17.47 (43.77%)|
> |MC|RB(800,1200)|33.89|31.42|31.49 (2.83%)|31.75 (13.36%)|31.90 (19.43%)|32.07 (26.31%)|32.11 (27.94%)|32.12 (28.34%)|32.14 (29.15%)|
> |MDS|RB(200,300)|27.89|28.61|28.56 (6.94%)|28.42 (26.39%)|28.28 (45.83%)|28.24 (51.39%)|28.18 (59.72%)|28.18 (59.72%)|28.19 (58.33%)|
> |MDS|RB(800,1200)|103.80|110.28|110.04 (3.70%)|109.12 (17.90%)|108.68 (24.69%)|108.34 (29.94%)|108.22 (31.79%)|108.17 (32.56%)|108.19 (32.25%)|
>
> The **Ours-1** configuration, which does not leverage cross-domain information transfer, yields a modest performance gain of approximately 5%. As the number of pre-training domains increases, the gains grow significantly, demonstrating the clear benefit of cross-domain knowledge transfer. Additional details, including the specific domains used for pre-training in each configuration for each downstream task, can be found in **Appendix G.6**.
>
>
> -----
>
> ## **Thank you again for your attention!**
>
> We sincerely hope that our clarifications and the additional experimental results adequately address your concerns. Your valuable feedback has been instrumental in helping us improve the clarity and completeness of our work. Should you have any further questions or require additional clarification, we would be more than happy to provide it. Thank you once again for your thoughtful review.

---

> ### Author Response · Authors · 2025-11-25
> **Looking forward to your feedback!**
>
> Dear Reviewer ZxEz,
>
> Thank you again for your thoughtful review and for the time you've dedicated to our work. We have added clarifications and experiments to resolve your concerns. As the discussion period comes to a close, we wanted to follow up and see if our responses have fully addressed your concerns. We are looking forward to hearing from you.
>
> Thank you again for your attention.
>
> Best regards,
>
> Authors

---

### Comment · Area_Chair_ze4y · 2025-11-24
**Discussion Period**

Dear reviewers,

The discussion period is now open. Please use the “Official Comments” to engage in discussions about each other's reviews and the authors' rebuttal, and update your assessments or comments as appropriate.

Did the authors' rebuttal adequately address your concerns? We kindly ask that you update your reviews based on these discussions and your evaluation of the rebuttal, even if your overall assessment remains unchanged.

Thank you all for your contributions.

Best regards, AC

---

### Author Response · Authors · 2025-11-30
**To Area Chair: Summary of Key Rebuttal Arguments and Revisions (Part 2/2)**

## **To Reviewer QCGp:**
1. **Scalability**: Addressed diminishing gains on large graphs by showing that pre-training on instances with only up to 25 nodes sustains substantial performance improvements (~20%) on very large-scale problems (e.g., ER graphs with 9000-11000 nodes).
2. **Baseline Comparison (GOAL, UNICO)**: Added empirical comparisons against GOAL and UNICO on their respective tasks (ATSP, 2DTSP, MVC, MIS). Results showed that our method consistently outperforms both.
3. **Warm-start Ablation**: Provided an ablation study showing that removing the warm-start phase causes a noticeable performance drop (Overall accuracy: 73.9% vs. 69.8%), confirming its importance for stabilizing learning.
4. **False Negatives**: Corrected the misunderstanding. Our data generation process makes false negatives highly improbable, and the modified loss experiment shows the impact is minimal.
5. **Temperature Selection**: Conducted a sensitivity analysis across a range of temperature values, showing that while temperature affects convergence speed, the final performance remains stable.
6. **SAT Model Failure**: Explained that the generalization failure on hard instances is attributed to a representational bias introduced by the GCN SAT backbone. Added analysis and experiments to show that the SAT model is still effective for in-distribution tasks and cross-model knowledge transfer, and single-domain fine-tuning successfully mitigates this bias.

## **To Reviewer nnWe:**
1. **Method Clarification**: Provided a detailed explanation of the pipeline from CO problems to GDPs to deterministic SAT graph transformation. Elaborated on the training paradigm, emphasizing how contrastive learning within the canonical SAT space forces the model to learn fundamental, transferable features.
2. **Methodological Contribution**: Corrected the misunderstanding, emphasizing that our core contribution is a novel training paradigm for representation learning, which is architecture-agnostic and fundamentally different from most existing works focused on developing specialized solving algorithms.
3. **Computational Complexity**: The concern about computational complexity is a misunderstanding. For training, there is only a one-time, upfront cost for pre-training the SAT model, which is the only procedure involving training multiple models. When applying to downstream tasks—including unseen problems—only an efficient alignment step with the fixed, reusable SAT model is required, not a new full training cycle. For inference, our approach introduces no additional computational overhead.
4. **No Available Code**: The claim of unavailable code is incorrect. Our initial submission included the pre-training code in the supplementary materials. We have further provided the pre-trained model checkpoint and downstream task code to ensure reproducibility of experimental results.

## **Main Revisions Implemented in the Manuscript**
1. Corrected calculation errors **in Table 4**.
2. Added further study on the role of multiple domain transfer **in Appendix G.6**.
3. Added empirical comparison with two multi-task baselines **in Appendix G.7**.
4. Added experiments and analysis on large-scale downstream CO problems **in Appendix G.8**.
5. Added ablation study on warm start **in Appendix G.9**.
6. Added ablation study on temperature selection **in Appendix G.10**.

---

### Author Response · Authors · 2025-11-30
**To Area Chair: Summary of Key Rebuttal Arguments and Revisions (Part 1/2)**

For your convenience, we summarize the key arguments in our rebuttal for all reviewers, and the main revisions in our manuscript below:

## **To Reviewer ZxEz:**
1. **Definitions**: Explained that confidence intervals estimate performance uncertainty, and that ER, BA, and BB are random graph generators for benchmarking.
2. **Training on Hard Instances**: Explained that hard instances are used for generalizability measurement. Added new experiments training on hard-level instances, showing ConRep4CO improves overall accuracy from 65.0% to 70.5%. Further experiments demonstrated that pre-training on hard instances sustains performance gains (~20%) on very large-scale downstream tasks.
3. **Marginal Multi-Domain Gain**: Conducted new experiments showing that performance gains on downstream tasks (MVC, MIS, MC, MDS) increase significantly with the number of pre-training domains, demonstrating clear benefits of cross-domain knowledge transfer.

## **To Reviewer FRhi:**
1. **Deeper Mechanistic Analysis of Our Approach**: Explained that our approach maps diverse problems into a unified space where core features become explicit and accessible to the model, and enhances performance on individual task by effective information transfer.
2. **Application to Broader CO Problems**: Explained that our approach can apply to a wide range of foundational NP-hard CO problems valued by the CO community, and added experiments on two general TSP tasks, ATSP and 2DTSP, where our approach achieves competitive performance against established baselines.
3. **Warm-start Ablation**: Provided an ablation study showing that removing the warm-start phase causes a noticeable performance drop (Overall accuracy: 73.9% vs. 69.8%), confirming its importance for stabilizing learning.
4. **Pre-training Diversity**: The experiments on the number of pre-training domains (Ours-1 to Ours-7) were presented to investigate the influence of pre-training diversity. The results show that with more diversity, the gains grow significantly, and eventually saturate. We find that pre-training with 5-6 domains offers a favorable trade-off.
5. **Practical Scalability**: Explained that the SAT transformation possesses polynomial-time complexity and is not a bottleneck, and the training process for various downstream tasks is efficient after the one-time pre-training. Added experiments to show that our approach achieves substantial performance improvements (~20%) on very large-scale problems (e.g., ER graphs with 9000-11000 nodes) without introducing a large training burden.

---

### Author Response · Authors · 2025-11-30
**To Area Chair: Core Contributions of Our Paper**

The core contributions of our paper are summarized below:

1. **Novel Pre-training Paradigm**: We propose ConRep4CO, a contrastive pre-training framework that advances representation learning for Combinatorial Optimization (CO) by learning representations across diverse CO problems, moving beyond established domains like vision and text.
2. **Cross-domain Contrastive Learning Scheme**: We design an instance-level contrastive learning objective that aligns original CO problem instances with their canonical SAT-form counterparts, learning domain-invariant structural features and enabling effective cross-domain information transfer and mutual enhancement.
3. **Strong Empirical Results**: Extensive experiments demonstrate that ConRep4CO significantly improves representation quality, in- and cross-domain generalizability, and solution performance on various popular CO tasks, establishing it as an effective pre-training method for CO.

Reviewers also recognize our work as: "novel", "promising extensibility", "clear and easy to follow" (Reviewer ZxEz); "robust generalization", "a viable paradigm for unified representation learning in the CO domain" (Reviewer FRhi); "creative and theoretically grounded", "compelling and elegantly exploited" (Reviewer QCGp); "improve performance in most cases" (Reviewer nnWe).

---

### Author Response · Authors · 2025-11-30
**To Area Chair: Summary of Reviewer Actions during Discussion Phase**

Dear Area Chair,

Thank you for your tremendous effort and dedication to the community, especially under these exceptional circumstances. Your work is greatly appreciated.

For your convenience, we would like to provide a concise summary of the reviewers' score updates during the discussion phase prior to the recent reset.

We submitted our initial rebuttal on 18 Nov 2025 (UTC+0). Following that, we received the following responses:

* **On 20 Nov 2025, 19:23 (UTC+0)**, **Reviewer QCGp** acknowledged our rebuttal and **increased the score to 6**.

* **On 26 Nov 2025, 1:44 (UTC+0)**, **Reviewer ZxEz** acknowledged our rebuttal and **increased the score to 6**.

* **On 26 Nov 2025, 4:08 (UTC+0)**, **Reviewer FRhi** acknowledged our rebuttal and **maintained the score of 6**.

**Reviewer nnWe** did not participate in the discussion and maintained the initial score of 4.

**In summary, prior to 27 Nov 2025, the paper's ratings were 4, 6, 6, 6, resulting in an average score of 5.5**.

Thank you for your time and dedication to the community!

Best regards,

Authors

---

### Meta-Review · Area_Chair_RARx · 2026-01-09

**Summary:**

The reviewers raised several concerns about the paper. One common issue is the clarity of the methodology, like how graph decision problems relate to Boolean Satisfiability. Reviewers also suggest missing empirical comparisons with other related methods, such as GOAL and UniCO, to provide more baselines. Scalability of the proposed method is another concern, because the experiments primarily focus on small graphs, so there is a concern whether the proposed method can do well on larger, more complex instances typical of real-world combinatorial optimization problems. Also, there are questions about computational complexity, regarding the warm-start strategy and code reproducibility.

**Reviewer Concerns:**

The authors provided a rebuttal to the concerns, including explanations of the details and new experiments. Most of the concerns have been addressed. The paper's contribution has convinced most reviewers to give a positive rating. A new contrastive pre-training framework has been proposed to enhance representation learning for Combinatorial Optimization. Extensive experiments have been conducted to show that the proposed ConRep4CO has state-of-the-art performance across several CO tasks. Thus, the AC suggests acceptance of the work.

**Reviewer Scores:**

The reviewers would change their scores to 4,6,6,6 after the rebuttal. The reviewer nnWe did not participate in the discussion, but I believe this reviewer may also increase the score.

---

### Decision · Program_Chairs · 2026-01-26

Accept (Poster)